# Quantifying electronic and geometric effects on the activity of platinum catalysts for water-gas shift

Xiansheng Li[1,2,3,9], Xing Wang [2,4,9], Arik Beck [1,2], Mikalai Artsiusheuski [1,2], Qianyu Liu[5], Qiang Liu[2], Henrik Eliasson[3], Frank Krumeich [1], Ulrich Aschauer [4,6], Giovanni Pizzi [2,7,8], Rolf Erni [3], Jeroen A. van Bokhoven [1,2] ✉ & Luca Artiglia [2] ✉

The unique catalytic activity of small nanoparticles can be attributed to their distinctive electronic structure and/or their ability to expose sites with a unique geometry. Quantifying and distinguishing the contributions of these effects to catalytic performance presents a challenge, given the complexity arising from multiple influencing factors and the lack of a quantitative structure-activity relationship. Here, we show that the intrinsic activity of platinum atoms at the perimeter corner sites is three orders of magnitude higher as a result of an electronic structure effect, with a threshold occurring at an average nanoparticle size of 1-1.5 nm. The contributions to the activity of atomically dispersed platinum, large nanoparticles and sodium-induced support modification are minor. This comprehensive and quantitative structure-activity correlation was demonstrated and verified on real-world Pt/CeO$_2$ catalysts for the water-gas shift reaction by utilizing operando X-ray photoelectron spectroscopy, in situ scanning transmission electron microscopy, electron energy-loss spectroscopy, theoretical calculations, and kinetic models.

The activity of a supported metal heterogeneous catalyst is the sum of contributions from all active sites, which originate from individual nanoparticles (NPs) of different sizes[1]. A crucial aspect of structure-activity relationship research is to quantify the reactivity of these sites[1–3] and correlate it with variations in electronic structure[2,4–8] and morphology[9–12]. Yet, due to the complex interplay of factors that determine activity[5,13], such as size effects[1,3,9–12,14], support effects[9,12,15,16], promoters[17–20], and reaction conditions[21], drawing such a relationship is not trivial.

For instance, the water-gas shift (WGS) reaction[22], describing reaction (1)

$$CO + H_2O \rightleftharpoons CO_2 + H_2 \, (\Delta H = -41.1 \mathrm{kJ/mol}) \tag{1}$$

serving as a vital industrial process and a benchmark for fundamental catalysis research, is the subject of intense debate about the structure of active sites in noble metal catalysts[22–24]. In situ and

[1]Institute for Chemical and Bioengineering, ETH Zurich, Zurich, Switzerland. [2]Paul Scherrer Institute, Villigen, Switzerland. [3]Electron Microscopy Center, Empa – Swiss Federal Laboratories for Materials Science and Technology, Dübendorf, Switzerland. [4]Department of Chemistry, Biochemistry and Pharmaceutical Sciences, University of Bern, Bern, Switzerland. [5]Institute of Informatics, University of Zurich, Zurich, Switzerland. [6]Department of Chemistry and Physics of Materials, University of Salzburg, Salzburg, Austria. [7]Theory and Simulation of Materials (THEOS), École Polytechnique Fédérale de Lausanne, Lausanne, Switzerland. [8]National Centre for Computational Design and Discovery of Novel Materials (MARVEL), École Polytechnique Fédérale de Lausanne, Lausanne, Switzerland. [9]These authors contributed equally: Xiansheng Li, Xing Wang. ✉e-mail: jeroen.vanbokhoven@chem.ethz.ch; luca.artiglia@psi.ch

operando studies have revealed that NPs are potentially active entities[17,23,25–27], and WGS activity has demonstrated a marked dependence on NP size[11,12]. To date, the size effect of WGS has been ascribed to a geometric effect, with the assumption that the inherent activity of NPs is independent of their size[9,11,12]. However, as NPs shrink down to the nanometer scale, their distinct electronic structure due to surface relaxation[28], quantum size effects[3,28] and metal-support interactions[6,16] may lead to starkly different catalytic performance[1,29]. Furthermore, when the metal becomes atomically dispersed, the unique atom utilization and electronic structure render such species highly attractive[24]. Consequently, atomically dispersed (AD) species have been widely regarded as active sites for WGS since 2003[18,19,30,31]. Since AD species may coexist with NPs[32], it is difficult to disentangle their individual contributions to the activity, perpetuating the debate between geometric and electronic structure effects. Accurately quantifying the respective contributions of electronic structure and geometric effects to catalytic activity, as well as pinpointing the critical size at which they occur, poses a significant challenge[33]. Moreover, the introduction of promoters, such as alkali cations, further muddies the water: their interaction with the supported metal[18–20,31], the support[17,34,35], and the metal-support interface[36], and the resulting impact on WGS activity remains debated and unclear.

While researchers have endeavored to disentangle geometric and electronic structure effects by using size-tailored catalysts[9,13] or employing selective site-blocking strategies such as atomic layer deposition[5], coupled with theoretical calculation and electron microscopy techniques, limitations persist. Firstly, real-world catalysts present an added level of complexity, not per se covered by these "model" catalysts[37,38]; secondly, the metal-support interaction effect of the blocking depositions (e.g., Al₂O₃ and FeOₓ) cannot be eliminated[5,39]; and thirdly, ex-situ techniques provide limited information[21,38,40]. To address these limitations, operando ambient pressure X-ray photoelectron spectroscopy (APXPS) serves as an ideal tool[41], capable of identifying metal atoms with varying coordination environments[42], electronic structures[6], and the chemical state of support[43] together with catalytic activity under working conditions.

In this work, we quantify the electronic structure effects on WGS activity and identify platinum active sites on sodium-modified Pt/CeO₂ powder catalysts under relevant WGS conditions by combining operando XPS, in situ and ex situ scanning transmission electron microscopy(STEM), electron energy-loss spectroscopy(EELS), theoretical calculations, and quantitative catalytic kinetic models. This approach excludes the contributions from AD species, large NPs, and sodium-induced support modification to the overall WGS activity. In this way, we present a comprehensive analysis of the structure-activity relationship, taking into account a diverse array of factors. The intrinsic activity of corner platinum sites is boosted by 1380 times as a result of the electronic structure effect, with a threshold occurring at an average NP size of 1–1.5 nm.

## Results and discussion
### Reactivity of AD and NP species towards WGS
A Pt/CeO₂ powder catalyst, containing 1 wt% AD platinum, was synthesized by atom trapping[44], referred to hereafter as AD_0 (see details in the "Methods" section and in Supplementary Table 1). Figure 1a shows the WGS light-off curves for AD_0 in 100 ppm CO and 300 ppm H₂O during the first conversion heating and cooling plots followed by a second cycle. Such conditions are industrially relevant for hydrogen fuel cell applications, where ppm levels of carbon monoxide are sufficient to poison the platinum catalyst[45]. No activity is detected in the first cycle during the early ramp-up stages. At 300 °C, the CO conversion spikes to about 70% and then quickly reaches thermodynamic equilibrium conversion at higher temperatures. During cooling from 300 to 100 °C, the activity is higher than during the ramp-up process, suggesting that the catalyst is activated through structural evolution at

around 300 °C. The second cycle of light-off experiments shows the same performance as that measured during the cooling of the initial cycle, suggesting that the activation process is complete by the time temperature reached 500 °C during the first ramp. High-angle annular dark-field (HAADF) STEM images (Fig. 1b, c and Supplementary Fig. 1) of AD_0 after the fast-WGS test (see details in the "Methods" section) reveal that, in the AD_0, no platinum NPs are present within the imaged area; only AD species are identified. In contrast, aggregated NPs with an average size of ~1.5 nm are detected on the sample after WGS. The ramp-up and cool-down processes were then reproduced during operando XPS measurements, at partial pressures of the reactants (0.1 mbar CO + 0.3 mbar H₂O; see details in the "Methods" section), matching those in the gas feed of the flow reactor. Figure 1d displays the photoemission spectra and peak fittings of the Pt 4f region. Four components, labeled a, b, c, and d are shown in the Pt 4f₇/₂ spectral range. Based on prior in situ studies of model platinum surfaces[6,42] and AD catalysts[46,47], we assign these components to platinum atoms in the bulk (71.0 eV, a), on terraces (71.6 eV, b), low-coordinated atoms on NPs (72.2 eV, c), and AD Pt²⁺ (72.8 eV, d). Such assignment is also in line with the binding energies (BE) measured on platinum NPs supported on CeO₂[6,48]. The observed BE of different metallic platinum species on NPs (a, b and c) are affected by final state effects, electronic metal–support interactions (EMSI) and Pt-CO interactions. On the one hand, when the metal-metal coordination number of a platinum atom decreases, lower screening of the core-hole leads to an increase in BE[49]. On the other hand, the electron transfer from platinum to the ceria support leads to partially charged Pt NPs and the partial reduction of Ce⁴⁺ to Ce³⁺, which contributes to the higher BE of Pt 4f[6]. In addition, CO adsorption can both induce the creation of under-coordinated structures and form back-donation bonding states between the Pt molecular orbital and the 2π orbital of CO, altering the charge density of platinum[42,50]. However, it is important to note that, according to density functional theory (DFT) calculations (see the "Methods" section for more details), the observed shift in binding energy is smaller in the case of small Pt clusters compared to flat Pt surfaces. At 100 °C and 0% CO conversion, the Pt 4f spectrum can be fitted by a single symmetric component centered at 72.8 eV (4f₇/₂), corresponding to AD species (i.e., Pt²⁺). This agrees with the characteristic STEM image of AD species shown in Fig. 1b and Supplementary Fig.S1. No change in the Pt 4f spectrum occurs upon a temperature increase to 250 °C. At 300 °C, the originally symmetric Pt 4f spectrum becomes asymmetric, and can no longer be adequately fitted using a single-component model. This necessitates the introduction of an additional low-BE component, which corresponds to metallic Pt⁰ nanoparticles formed upon partial reduction of Pt²⁺. This co-existence of AD and NP species is confirmed by the post-reaction STEM observations in Supplementary Fig. 2. In the course that AD platinum species strongly interact with oxygen on ceria, the reduction of AD Pt²⁺ implies sintering, as AD Pt⁰ atoms are less stable than Pt⁰ NPs[47,51]. Although previous studies reported the existence of reduced single atoms on specially prepared CeO₂ islands[52], our powder samples do not employ such a synthesis strategy, making the occurrence of CeO₂ islands with isolated platinum atoms and ions highly unlikely. Therefore, the reduction of Pt²⁺ is accompanied by the sintering of AD species into metallic NPs. Bulk and terrace metal sites indicate the formation of clusters or NPs, while low-coordinated atoms are associated with small nanoclusters and to edge and corner sites on larger NPs. During subsequent cooling to 100 °C, the newly formed Pt⁰ species do not oxidize back to AD Pt²⁺, in agreement with STEM images showing ~1.5 nm NPs on AD_0 after WGS. The quantitative evolution of the Pt 4f species at various temperatures in the WGS environment is plotted in Fig. 1e. The hydrogen production (in blue), monitored during the APXPS experiment, and the CO conversion (in red), measured in the flow reactor, are also displayed. The activity of the catalyst in both experiments is similar, allowing a correlation with the APXPS results. The onset of activity at 300 °C in both

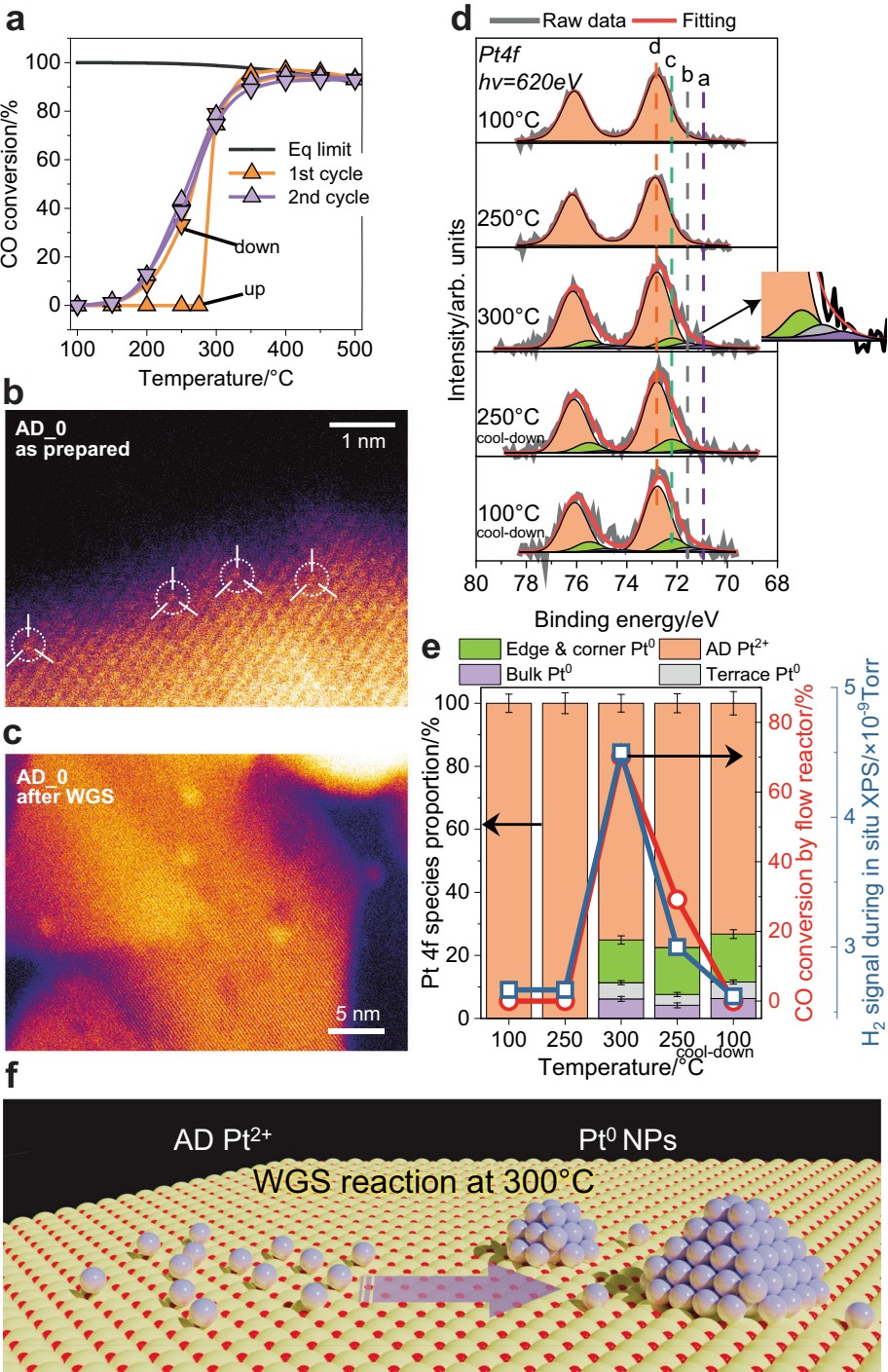

**Fig. 1 | Activation of atomically dispersed Pt/CeO₂ via Pt sintering, as demonstrated by catalytic testing, STEM, and operando XPS. a** Light-off curves of AD_0 sample. "Eq limit" represents the WGS thermodynamic equilibrium limitation. Feeding: 100 ppm CO + 300 ppm $H_2O$, total flow=100 ml/min. HAADF-STEM images of AD_0 samples as prepared (**b**) and after WGS (**c**). **d** Experimental XP spectra and peak fitting of the Pt $4f$ region obtained in 0.1 mbar CO + 0.3 mbar $H_2O$, during ramp-up at 100–300 °C and cool-down at 300–100 °C, using a photon energy of 620 eV. The Pt $4f$ region contains four components corresponding to bulk atoms (a), terrace atoms with adsorbed CO (b), low-coordinated atoms with adsorbed CO (c) and atomically dispersed (AD) $Pt^{2+}$ ions (d). **e** Stacked bar plot showing the evolution of Pt $4f$ species on AD_0 at different temperatures; line plot showing the CO conversion acquired by flow reactor in 100 ppm CO + 300 ppm $H_2O$ (red axis) and $H_2$ signal recorded during the APXPS measurements by the residual gas analyzer (RGA) mass spectrometer (blue axis). **f** Schematic diagram of the structural evolution of the AD platinum in WGS at 300 °C.

experiments relates to the reduction of platinum. In the flow reactor, AD_0 is not active for WGS at 250 °C during the first ramp-up; however, after activation during cooling, the sample shows a rate of 30.5 mmol$_{CO}$·mol$_{Pt}^{-1}$·s$^{-1}$ (Supplementary Fig. 3). The sintering of AD platinum into NPs activates the catalyst towards the low-temperature WGS reaction (Fig. 1f).

## Size dependence of WGS over Pt/CeO₂

Another Pt/CeO₂ sample initially containing platinum NPs (NP_0) was synthesized (see the "Methods" section and Supplementary Table 1) and tested to allow comparison with AD_0. Supplementary Fig. 4 shows the WGS activity of AD_0 and NP_0. The ramp-up behavior of NP_0 is similar to that of AD_0 after the first light-off cycle, supporting

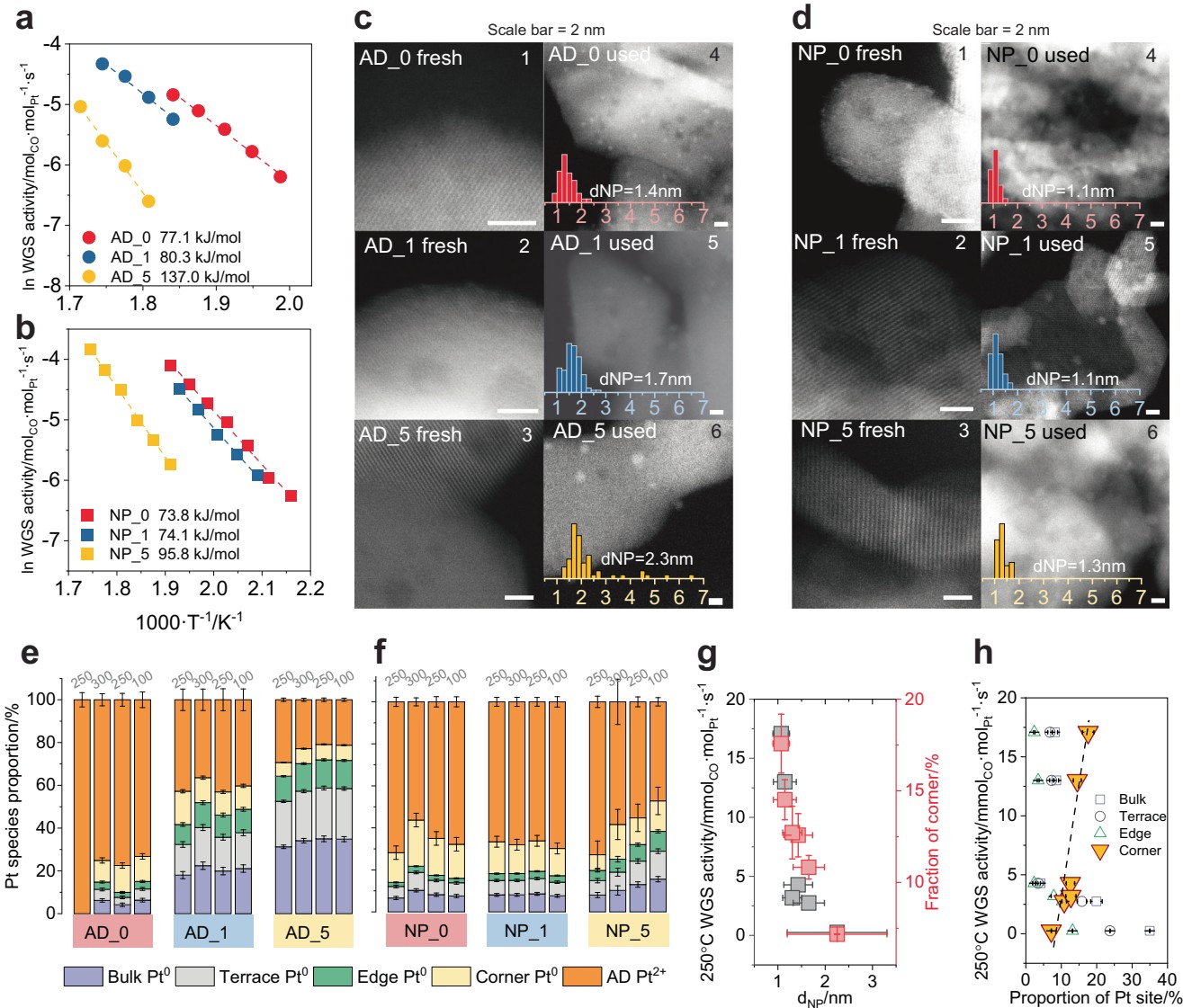

**Fig. 2 | Size-dependent WGS activity and Pt structural characteristics, and their direct correlation.** Arrhenius plot for WGS obtained on (**a**) AD_x and (**b**) NP_x samples. The corresponding activation energies are also noted. Feeding: 100 ppm CO + 300 ppm $H_2O$ + 500 ppm $H_2$ + 100 ppm $CO_2$, total flow=100 ml/min. HAADF-STEM images of (**c**) AD_x and (**d**) NP_x catalysts loaded with various amounts of sodium (c1−c3 & d1−d3: fresh samples before WGS; c4−c6 & d4−d6: used samples after WGS). The scale bar in the STEM images is 2 nm. The size distribution histograms of the used samples are shown. Stacked bar plot showing the structural evolution of *Pt 4f* species on (**e**) AD_x and (**f**) NP_x samples at different temperatures. (**g**) Size dependence plot of WGS activity at 250 °C(in black) and fraction of corner sites by XPS acquired at 250 °C during cool-down (in red). **h** Direct correlation between WGS activity and the fraction of different platinum sites (bulk, terrace, edge, and corner).

the hypothesis of activity booster due to partial sintering of AD platinum.

Alkali metals are commonly employed as catalyst promoters in industry[53,54], with particular emphasis on their utilization in WGS catalysts, where the catalyst performance is enhanced by stabilizing the dispersion of the supported noble metals[18–20,31,55]. To investigate the effects of alkali modification, we added different amounts of sodium to the AD_0 and NP_0 catalysts (denoted as AD_x and NP_x, respectively, with x representing the atomic ratio of Na to Pt (0, 1, 5)). Supplementary Fig. 5 shows the light-off curves for the AD_x samples. As more sodium is introduced, the AD_x catalysts exhibit decreased activity in both the ramp-up and ramp-down curves. A pattern consistent with the light-off curves is observed in the Arrhenius plots in Fig. 2a. The WGS activity at 250 °C falls by a factor of 17 as the sodium loading x increases from 0 to 5. Simultaneously, the apparent activation energy ($E_a$) over the catalysts increases from 77 to 138 kJ/mol. The effect of sodium addition is similar for both AD_x and NP_x catalysts: the CO

conversion curves gradually shift towards higher temperatures with increasing sodium loading in the light-off experiments (Supplementary Fig. 6). From the Arrhenius measurement (Fig. 2b), an increase in $E_a$ is observed. In Fig. 2, panels c1 to c3 show the morphology of the fresh AD_x catalyst, while panels c4 to c6 show the morphology of the corresponding samples after the WGS reaction. Supplementary Figs. 7 and 8 show the representative STEM images and Supplementary Table 2 lists the counted platinum NP average diameters. The addition of sodium does not induce an evident change in morphology of AD_x upon calcination in air. Platinum on the surface of the fresh AD_x catalysts remains highly dispersed and neither nanoparticles nor atom assemblies are observed. On the AD_x and NP_x samples after WGS test, however, platinum agglomerates into NPs, and the more sodium is added, the larger the NPs become. Histogram of size distribution given in panels c4 to c6, d4 to d6 as well as in Supplementary Fig. 9 reveal that platinum NPs increase in size, on average from 1.4 nm for AD_0 to 2.3 nm for AD_5. The image of the fresh NP_0 sample (Fig. 2,

panel d1) displays several platinum assemblies. After addition of sodium and calcination in air, platinum becomes atomically dispersed with no evidence of nanoparticles on NP_1 and NP_5, suggesting that sodium promotes the redispersion of platinum from NPs. Such an effect is consistent with observations in other systems studied by Flytzani-Stephanopoulos et al. and other groups in the field[18,19,31,55]. Theoretical calculations of the effect of sodium addition (in the "Methods", S1.3 and Supplementary Fig. 10) show that NaOH promotes metal redispersion on the oxide surface by stabilizing cationic AD platinum species[18,19,51]. Like AD_0 after the WGS reaction, platinum NPs exist on NP_x samples (Fig. 2, panels d5 and d6). The NP diameter shows a slight increase with increasing sodium content. The morphological characterization of both the AD_x and NP_x series shows that sodium promotes the agglomeration of platinum into NPs under WGS conditions and that higher sodium loadings lead to larger particles. Figure 2e, f display the quantitative results of operando XPS investigations of Pt $4f$ for the AD_x and NP_x samples, reproducing the protocol described in Fig. 1d. Supplementary Fig. 11 presents the raw Pt $4f$ spectra at 250 °C during cool-down, illustrating significant changes in Pt $4f$ as sodium increases, accompanied by a rise in low BE species, which signifies enhanced sintering.

To demonstrate these structural changes quantitatively, we adopted the geometric effect in the peak fitting of Pt $4f$, based on morphology as a fitting parameter guide, as detailed in S1.1 and Supplementary Fig. 12. Following this strategy, we fixed the ratio of each component (bulk, terrace, edge, and corner sites) of the NP at specific sizes. It is worth mentioning that platinum NPs exhibit a wide range of shapes owing to variations in size and interaction strengths with different supports[56–58]. However, obtaining statistically significant shape distributions proves challenging due to various factors, including the reaction atmosphere, beam-induced effects, image resolution, fluxional nanoparticle structure, etc. Taking all these factors into account, we selected the truncated octahedral shape as a meaningful and reliable representation based on theoretical calculations and in situ STEM evidence (a more comprehensive discussion refers to section S1.2, Supplementary Movies 1–3 and Supplementary Figs. 13, 14). Supplementary Fig. 15 provides a detailed example of the correlation between the fitting of the components and the NP sites, and Supplementary Figs. 16, 17 and Supplementary Tables 3–5 display the results obtained and fitting parameters used for the Pt/CeO$_2$ catalysts. By means of this strategy, we successfully separated bulk, terrace, edge and corner sites on NPs and AD Pt$^{2+}$, obtaining satisfactory correlations between the deconvoluted components and raw data (see Supplementary Fig. 18 and Supplementary Table 6). The results confirm that, on all catalysts, AD platinum gradually converts into NPs under WGS conditions. Furthermore, the structural evolution correlates with the sodium loading: the more sodium added, the higher the fraction of Pt$^0$ NPs (bulk, terrace, edge and corner), at the cost of AD Pt$^{2+}$. The integration of XPS and STEM enables a comprehensive evaluation of the composition of sites present: XPS discriminates between AD and NP species while STEM provides the structural information on NPs to further deconvolute the number of particular Pt sites within them.

Figure 2g displays the WGS rate and the corner fraction as a function of the diameter of the Pt NPs (d$_{NP}$). Consistent with reported results[11], the WGS activity at 250 °C decreases with increasing d$_{NP}$. The activity plot can be well fitted by the size dependence of the corner sites, highlighting their significance. In the literature, such an overlap has supported the assertion that WGS size dependence is attributed to the geometric effect, and corner sites maintain identical activity regardless whether they exist in small or large NPs[9,11,12]. However, Fig. 2h shows that when the WGS activity is correlated with the fraction of corner sites, the relationship is not proportional. As an example, the AD_5 sample, which has a fraction of ~7% corner sites, displays a WGS rate of 0.00025 mol$_{CO}$·mol$_{Pt}$$^{-1}$·s$^{-1}$, whereas the rate rises to 0.017 mol$_{CO}$·mol$_{Pt}$$^{-1}$·s$^{-1}$ in the most active NP_0, which has 18% corner sites.

Two potential explanations for this discrepancy are: i) the fraction of corner sites extrapolated from the models does not represent actual NPs and ii) the corner sites are not all equally active, suggesting that an additional electronic structure effect plays a role. Considering the first possibility, models of NP with different shapes, depicting stronger and weaker interactions between platinum and ceria, were tested to reproduce a faster decrease in corner sites with increasing NP size. None of the models was able to alter the trend in the corner sites as drastically as observed in experiments, as discussed in S1.4 and depicted in Supplementary Fig. 19. Such results suggest the coexistence of corner sites with various activities.

## Quantitative structure-activity relationship

In this section, we considered electronic structure effects to estimate the activity of different types of corner sites. As the size of supported NPs decreases to the nanometer range, the distinct electronic structure of small NPs at specific sizes enables unique reactivity[6,28]. In the case of real-world catalysts, such as those examined in this study, it is common for the size distribution of the supported NPs to exhibit a degree of broadness[59,60]. Consequently, catalysts with larger average NP sizes tend to contain very small NPs as well, and the quantity of these small NPs could play a crucial role in determining the overall catalytic performance. In this context, we categorize the NPs as "small" and "large" according to their size, and the thresholds are set at 1.2, 1.25, 1.3, 1.4 and 1.5 nm. Subsequently, the fraction of small NPs (Supplementary Table 7) is used to count the corner sites on small NPs with potentially high catalytic activity. Supplementary Fig. 20 shows the correlations between WGS activity and the fraction of corner sites on small NPs for different thresholds. The plots demonstrate a notable proportional relationship, with the most optimal linear correlation observed at thresholds of 1.2, 1.25, and 1.3 nm. This observation underscores a remarkable reactivity of small NPs when their size decreases below a certain threshold. This observation aligns with the findings of Lykhach et al. [6], who reported that the maximum charge transfer per platinum atom to ceria occurs with particle sizes between 1 and 1.5 nm. Rodriguez et al. [48] also found that the highest WGS activity is attained at 0.2 ML Pt/CeO$_2$(111), with an approximate NP diameter of 1.7 nm. Additionally, Nørskov et al. [28] reported that electronic finite-size effects in platinum clusters start to manifest at sizes below 1.6 nm, where the CO adsorption energy becomes more negative. Therefore, the outstanding activity displayed by these small NPs can be potentially ascribed to their pronounced interaction with the support and/or unique binding with reactive intermediates due to electronic structure effects. Nevertheless, the fractions of bulk, terrace, and edge sites on small NPs are also altered and might display a correlation with WGS activity. For this reason, we also evaluated the contribution of those sites on small NPs (<1.25 nm) to the overall activity, by means of multi-linear regression. Supplementary Table 8 provides details of the algorithm and results, which do not display significant contributions.

Along with platinum, the support is also important for the WGS reaction, and the metal-support interface has been proposed to be actively involved in the reaction[25,61,62]. We investigated Ce $3d$ spectra in Section S1.5, as well as in Supplementary Figs. 21–23 and Supplementary Table 9. Ce $3d$ spectra of AD_x and NP_x samples exhibit relevant modifications at different temperatures under WGS working conditions, suggesting that WGS involves the participation of the support. For this reason, platinum atoms in contact with ceria (perimeter sites[9]) are critical. Supplementary Fig. 24 illustrates the good correlation of WGS activity with the fraction of perimeter atoms on small clusters. However, both edge and corner sites are located at the perimeter between NPs and ceria. In order to disentangle their respective reactivity, we divided perimeter sites into corner@perimeter*small and edge@perimeter*small. Supplementary Fig. 24 demonstrates that only corner@perimeter*small sites correlate well with the activity.

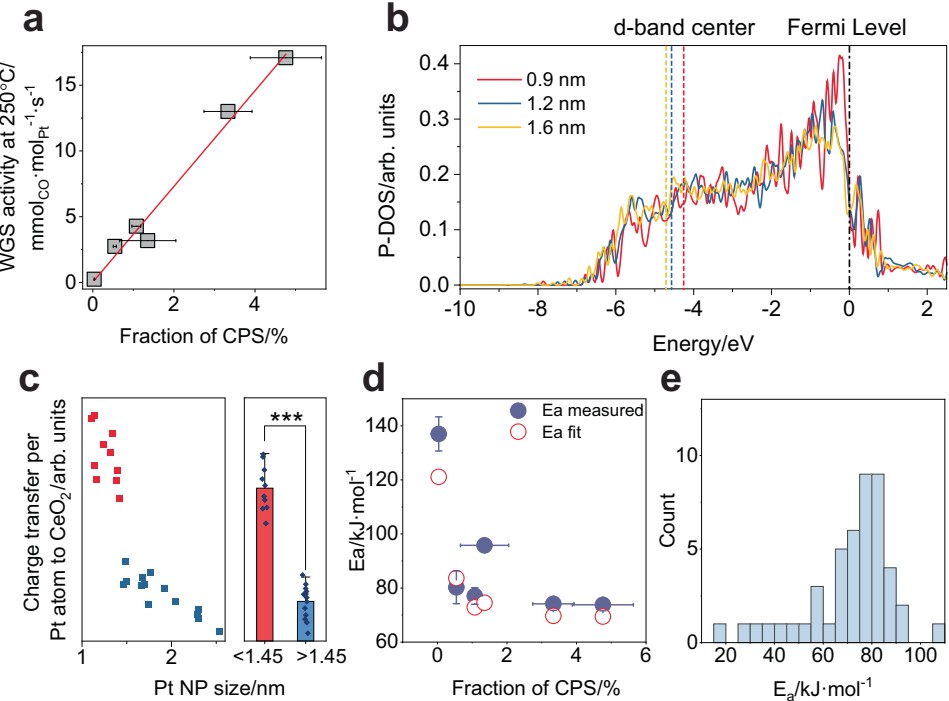

**Fig. 3 | Size-dependent electronic structure of Pt-CeO₂ interface and the role of corner-perimeter sites at smaller Pt particles in WGS activity. a** Direct correlation of WGS activity at 250 °C with the fraction of CPS. **b** Calculated projected density of states (P-DOS) for platinum NPs supported on a CeO₂ (111) surface, for three different NP sizes. The black vertical line indicates the Fermi level, set at zero energy. The three vertical dashed lines indicate the position of the center of the occupied *d* states. **c** Size dependence of averaged charge transfer per platinum atom to CeO₂ at the Pt-CeO₂ interface and the statistical analysis of group differences between NP size smaller, respectively larger than 1.45 nm. Significance is indicated as \*\*\**P* < 0.001 by Mann-Whitney *U* test. **d** Activation energy measured by experiment and estimated by kinetic model as a function of CPS fraction. **e** Previously reported activation energy distribution for WGS reaction over platinum catalysts.

Furthermore, it is necessary to consider how electronic structure effects influence Pt *4f* spectra: a shift towards higher BE is counted on smaller NPs, particularly those smaller than 1.25 nm. For instance, the peak of terrace sites on small NPs (< 1.25 nm) might shift to higher BE, i.e. from 71.6 to 72.0 eV[6]. For edge/corner sites, the peak would also shift to higher energies (slightly higher than 72.2 eV but lower than 72.8 eV for AD species). Given the resolution of Pt *4f* XPS measurements in this work (~ 0.3 eV) and the BE of small clusters on model Pt/CeO₂[6], we assigned the peak at 72.2 eV to corner and edge sites, regardless of the NP size and of the potential BE shift of terrace atoms on small NPs. Supplementary Table 7 lists the average diameter of small NPs (< 1.25 nm) for all the samples and the corresponding fraction of corner, edge and terrace atoms. The fraction of corner sites on small NPs is then corrected by removing the terrace*small contribution from the peak at 72.2 eV. The comparisons of fitting before and after terrace peak correction are shown in S1.6, suggesting that the correction has minimal influence on the data processing and overall results. The corrected fraction of corner@perimeter*small is then named as CPS. Figure 3a plots the WGS activity as a function of the fraction of CPS, revealing a proportional relationship. The strong correlation implies that platinum species on larger nanoparticles, even if being corner sites, are less active[14]. The WGS activity stems from CPS, due to their distinct electronic structure[6,63]. This finding reveals that the size-dependent effect of Pt/CeO₂ in the WGS reaction arises from two key factors: (1) an increased number of interfacial corner sites, and (2) an intrinsic activity enhancement of these sites when the particle size falls below a critical threshold.

To quantify the intrinsic activity difference between CPS and other Pt sites, a non-linear Arrhenius kinetic model was developed and described in section S1.6, Supplementary Figs. 25–28 and Supplementary Tables 10 to 12. It incorporates all the kinetic data and

structural information of platinum and supports the proposed structure-activity relationship. Experimentally measured activity data are in good agreement with the kinetic model. According to the fitting, the estimated WGS rate for the CPS at 250 °C in 100 ppm CO + 300 ppm H₂O + 500 ppm H₂ + 100 ppm CO₂ is 0.34 mol$_{CO}$·mol$_{CPS}^{-1}$·s$^{-1}$. The value for all other sites is 0.00025 mol$_{CO}$·mol$_{site}^{-1}$·s$^{-1}$, i.e., a difference of three orders of magnitude. The intrinsic WGS activity of corner atoms at 250 °C in relation to platinum NP size is depicted in Supplementary Fig. 29. It displays a 1380-fold difference for small and large NPs when subjected to a strong alteration in electronic structure. Conversely, the activity remains constant in the case of mere geometric effect, irrespective of the NP size. This illustration demonstrates the substantial distinction between the widely recognized geometric effect and the electronic structure effect quantified in this study. The substantial difference in the intrinsic activity between larger and smaller NPs is corroborated by theoretical calculations of the reaction energy for the rate-determining step (RDS), which is typically considered to be the dissociation of the carboxyl in the associative mechanism of WGS[61,64]. As discussed further in the Methods and S1.7 and illustrated in Supplementary Figs. 30–33, the corner atoms on the 0.9 nm and 1.6 nm Pt NP models exhibit a difference of approximately 0.3 eV in reaction energy, demonstrating substantial difference in the activation barrier from their scaling relation[65,66]. This provides strong evidence for the pronounced disparity in the intrinsic activity of NPs of varied size.

In order to provide further support regarding the strong change in electronic structure as a function of NPs size, we conducted both theoretical and experimental analyses by calculating Bader charge and projected density of states (P-DOS), as well as by collecting EELS data on the model Pt/CeO₂ catalyst. P-DOS and Bader charge were calculated on three platinum NP models at sizes of 0.9, 1.2 and 1.6 nm. As

shown in Fig. 3b, a pronounced increase in the density of electronic states near the Fermi level is discernible for the smaller NP, with a distinct peak slightly below the Fermi level visible exclusively for the smallest NP (0.9 nm). This observation indicates an increased tendency of the $d$ electrons to engage in catalytic reactions, aligning with the expected sharp increase in catalytic activity for smaller particles. We also calculated, and indicated with vertical dashed lines, the centers of the occupied $d$ states, which are widely treated as a descriptor of catalysis[67]. These values also underscore a contrast between small Pt NPs (0.9 nm) and larger NPs (1.2 nm and 1.6 nm), indicating a difference in the catalytic performance of small versus larger NPs. Moreover, Bader charge analysis of perimeter atoms on the three NPs, as detailed in S1.8, Supplementary Fig. 34, and Supplementary Table 14, highlights a significant difference in charge transfer from platinum to ceria between small Pt NPs (0.9 nm) and larger NPs (1.2 nm and 1.6 nm), verifying the P-DOS and $d$-state center analysis. In addition to theoretical calculations, a STEM-EELS analysis was conducted to investigate the influence of Pt NP size on the oxidation state of $CeO_2$ nearby. Such value accounts for an average charge transfer from a Pt atom to $CeO_2$ at the Pt-$CeO_2$ interface. Detailed information on the sample preparation and the EELS experiments can be found in section S1.9, Supplementary Figs. 35-37, and Supplementary Table 15. As depicted in Fig. 3c, the average charge transfer from individual Pt atoms to $CeO_2$ decreases as the size of the Pt NPs increases. Moreover, a sharp decrease in charge transfer is observed at approximately 1.5 nm. Notably, there is a statistically significant distinction ($p$ value < 0.001) in charge transfer between small and larger NPs. Such result aligns well with the theoretical calculations and previous studies reported in the literature[6], which employed scanning tunneling microscopy and valence band photoelectron spectroscopy. These observations suggest that when the size of Pt NPs falls below the threshold of 1–1.5 nm, the charge transfer from individual Pt atoms to $CeO_2$ experiences a significant increase, resulting in a notable alteration of the electronic structure ($d$-state) of Pt NPs. This finding provides a validated and reliable explanation for the electronic structure effect of Pt/$CeO_2$ catalysts.

The threshold of 1.25 nm, determined through operando XPS and kinetic model displays discrepancies compared to STEM-EELS and theoretical calculations. Such deviation can be traced back to the errors arising from size distribution and shape heterogeneity of platinum NPs, the diverse $CeO_2$ facets present at the interface and the gap between model system and real-world catalysis. As highlighted in Supplementary Fig. 29, when the size of Pt NPs decreases below a threshold, typically falling within the 1–1.5 nm range, there is a notable enhancement in the intrinsic catalytic activity of Pt NPs towards the WGS reaction. This phenomenon is attributed to the significant alteration in the electronic structure impact of platinum induced by the distinct $d$ electronic states and/or larger Pt-$CeO_2$ charge transfer.

The kinetic model also provides insights into the variation of activation energy among different samples. It is considered that the apparent activation energy is a weighted sum of the individual activation energy of each site with respect their contribution to overall reaction rate. This relationship is illustrated in Fig. 3d, where it is evident that the obtained activation energy values align well with the kinetic model. Supplementary Table 16 lists the WGS activation energies of platinum-based catalysts reported in the literature since 1981; their distribution is shown in Fig. 3e. The activation energies were obtained from studies using various synthesis methods and platinum concentrations. Most of the values are distributed around 80 kJ/mol, in agreement with both our empirical data and the simulation in the case of Pt NPs of 1–2 nm.

The use of alkali metals as promoters is a common industrial practice. However, whether the role of alkali metals stems from modifications to the metal or the support remains controversial[17–20,31,34,35]. The typical inhibition effect of Pt/$CeO_2$ by sodium has been attributed to the

agglomeration of platinum into larger NPs under WGS working conditions (the size dependence of WGS is long established). To establish a comprehensive understanding of the structure-activity relationship, as well as to clarify the role of alkali metals, we address this topic in further detail in Section S1.10 and Supplementary Figs. 38–42. The interactions between alkali metals and the ceria support modify the reaction dynamics, resulting in a promotion effect due to the alleviation of $H_2$'s negative reaction order. Because of the dominant role of platinum sintering, we ultimately observe a predominating inhibitory effect on the overall catalytic performance. Therefore, we achieve a comprehensive analysis of the structure-activity relationship, taking into account contributions from AD species, low CN sites at small and large NPs and sodium-induced support modification. We also clarified potential compatibility issues between the operando APXPS experiments and catalytic tests in S1.11, Supplementary Figs. 43–45 and Supplementary Table 17. Our research findings are relevant in addressing discussions regarding the identification of active sites (Section S1.12) and are in line with the results acquired on model catalysts (Section S1.13).

In this work, we identified and quantified all platinum sites on Pt/$CeO_2$ under WGS working conditions by utilizing geometric and electronic structure effects to deconvolute Pt $4f$ photoemission peaks. The direct and quantitative relationship between the WGS activity and the site count allows for the distinction between primary and secondary contributions in the comprehensive structure-performance relationship study. Sodium promotes dispersion of platinum upon calcination in air and the sintering of platinum under WGS working conditions. The variation in electronic structure between small and large platinum NPs, with a threshold size identified at 1–1.5 nm, results in a discernible discrepancy in reactivity, which is estimated to be around 1380 times. The results obtained by operando XPS, and kinetic models can be well confirmed and justified by EELS, P-DOS and Bader charge calculations. Although ceria demonstrates an interaction with sodium, its contribution to the modification of WGS activity is minor. Such understandings are crucial for the rational design of WGS catalysts. For example, merely increasing the Pt loading (increasing the number of clusters and sites) without controlling the size does not boost the fraction of active sites.

Catalytic tests reproducing industrially relevant reaction conditions adopted during spectroscopic measurements, provide relevant information about the catalytic performance, thus allowing for an operando study and accounting for the pressure gap associated to APXPS[45,47]. Such a robust approach is generally applicable in real-world catalysts beyond model systems, to identify and quantify metal active sites. For example, CO oxidation, where pure geometric effects are insufficient to account for size dependence in small NPs[9] (< 2 nm) and specific sized NPs (~ 1.4 nm) are proposed to be more active[68], is a potentially suitable case study. The proposed methodology is essential for the advancement of catalytic science and technology, and for the development of sustainable and efficient industrial processes.

## Methods

### Material synthesis

The catalysts (1 wt% Pt/$CeO_2$) were prepared by incipient wetness impregnation (IWI). Supplementary Table 1 lists the designation and the methods of preparation of the samples in this work. The cerium(III) nitrate hexahydrate (99.9%, abcr Swiss) was heated in air at 350 °C for 2 h to obtain the bare ceria support. The required amount of the platinum precursor chloroplatinic acid, 99.9% (Alfa Aesar) for AD_0[44,69] and tetraammineplatinum(II) nitrate (Alfa Aesar) for NP_0[70,71] was dissolved in deionized (DI) water and the solution was added dropwise to the ceria during grinding. The powder was then dried overnight at 100 °C in air. After drying the powder for AD_0, it was heated to 800 °C at a rate of 5 °C/min, kept at 800 °C for 12 h, and cooled (1 °C/min) to 500 °C at then cooled further to room temperature (RT) in flowing air. The powder for NP_0 was calcined at 500 °C for 5 h in flowing air and then cooled to RT.

Sodium was added to the AD_0 and NP_0 samples by the IWI method. An appropriate amount of sodium nitrate was dissolved in DI water and the solution was added drop-wise to the fresh catalyst during grinding. The powder was then dried at 100 °C overnight, followed by calcination at 500 °C for 5 h in flowing air and then cooled.

## WGS performance

The WGS reaction was conducted in a Hiden CATLAB plug-flow reactor and the reaction products were analyzed by means of a Pfeiffer Omnistar quadrupole mass spectrometer. Two milligrams of catalyst, diluted with 0.2 g of inert SiC powder, were used to prevent temperature gradients. The main text mentions three types of reactions: light-off curve, fast measurements simulating conditions in an APXPS chamber and Arrhenius measurements. For the light-off curve measurements, the reaction temperature was increased from 50 °C to 500 °C and then decreased to 50 °C in steps of 50 °C. The flow mixture was 100 ppm CO, 300 ppm $H_2O$ and argon to balance, at a total flow rate of 100 ml/min in ambient pressure. This gas feed corresponds to partial pressures of 0.1 mbar CO and 0.3 mbar $H_2O$ during APXPS measurements. For fast measurements, the reaction temperatures were 100 °C, 250 °C, 300 °C, 250 °C and 100 °C, using the same catalyst mass and gas feed. For Arrhenius measurements, the gas feed was 100 ppm CO, 300 ppm $H_2O$, 500 ppm $H_2$, 100 ppm $CO_2$ and argon to balance, at a total flow rate of 100 ml/min. Samples used for Arrhenius measurements were initially treated in 100 ppm CO + 300 ppm $H_2O$ at 500 °C for 30 min for activation. The steady-state temperature steps are given in the main text. Water was dosed by means of a temperature-regulated saturator. CO was purified by flowing it through a copper mesh trap at 300 °C (to remove nickel carbonyls). DI water in the saturator was purified by boiling and bubbling helium through it for 1 h. The error bar of $E_a$ is the standard deviation during linear fitting.

The standard gas feed for reaction order test was the same to the Arrhenius test. We worked in the temperature region where the CO conversion is lower than 10% (kinetically controlled regime). In general, to determine the reaction order of a compound, its concentration is varied while keeping the concentrations of other components constant. Four concentrations were varied over the ranges 0.0053–0.0206 % CO, 0.002–0.01 % $CO_2$, 0.013–0.148 % $H_2O$, and 0.022–0.078% $H_2$. Three different temperatures (260, 270, 310 °C) were used for AD_0, AD_1 and AD_5 respectively.

## Ex situ STEM and EDXS

Scanning TEM (STEM) at 200 kV, with high-angle annular dark field (HAADF) imaging, was performed with a Hitachi HD 2700 electron microscope, equipped with a probe-corrector, at the Scientific Center for Optical and Electron Microscopy (ScopeM) of the ETH Zurich. Electron microscopy was done ex situ in vacuum ($10^{-6}$ mbar pressure range). Due to the rather small difference in the Z contrast of Pt and Ce, the detection of AD species in STEM is challenging compared to the detection of small NPs (~1 nm). Therefore, the absence of detectable NPs in STEM is often interpreted as the sole presence of AD Pt and small clusters. Energy dispersive X-ray spectroscopy (EDXS) was done at $U_{acc} = 200$ kV on a HD2700CS instrument (Hitachi; FEG) equipped with an EDX spectrometer (EDAX (Ametec)). The fresh and used samples were transported in common sample vials in air. The fresh samples were measured right after synthesis and the used samples were measured after the fast WGS measurements. The error bar is the standard deviation calculated from the data in Supplementary Table 2.

## APXPS

Ambient pressure XPS measurements were carried out at the X07DB In Situ Spectroscopy Beamline at the Swiss Light Source (SLS) synchrotron. The catalyst powders were ground and pressed (1.5 ton) onto a silver mesh. The silver mesh was fixed to a manipulator and introduced into the solid–gas interface endstation, which allows for precise dosing of gas/gas mixtures under flow conditions[72,73]. Water was introduced into the chamber from a saturator, the temperature of which was controlled to regulate the equilibrium water vapor pressure. Gas mixtures were dosed and controlled by means of mass flow controllers and pumped away with a tunable diaphragm valve connected to a root pump. This allows for the dosing of relevant gas flows and precise control of the pressure during the experiments. The pressure was monitored by means of Baratron measurement heads. The samples were heated using a tunable IR laser hitting the back of the sample holder, and the temperature was monitored by means of a Pt100 sensor. After alignment, the sample was investigated by acquiring all the photoemission peaks in sequence, while being exposed to a specific gas at different temperatures. The samples were exposed to a mixture of 0.1 mbar CO and 0.3 mbar $H_2O$ at 100, 250, 300, 250 and 100 °C. Linearly polarized light was used throughout the experiments. *Pt 4f* peaks were acquired with $h\nu = 620$ eV, corresponding to a kinetic energy of ~540 eV. This value was chosen to maintain a high energy resolution, and sufficient photon flux. At this kinetic energy, the mean escape depth (defined as $\lambda*\cos(\theta)$, where $\lambda$ is the inelastic mean free path of photoelectrons in the solid and $\theta = 30°$) of *Pt 4f* photoelectrons is ~7 Å. The binding energy (BE) scale was aligned by means of the reference *Ce 4d* peak acquired at the same excitation energy. During ramp-up at 100 °C, the spectra displayed differential charging; the results are not given to avoid possible misinterpretation. After subtraction of the Shirley background, the photoemission peaks were fitted by Voigt-shaped functions. An asymmetric Doniach-Sunjic line shape[74], the parameters of which were evaluated from the fitting of the reference platinum metallic foil, was used to deconvolute the bulk platinum peak[42]. *Ce 3d* spectra were collected under the same reaction conditions used for *Pt 4f*, but with an excitation energy of 1200 eV. The *Ce 3d* spectrum of $CeAlO_3$ was measured as a reference for $Ce^{3+}$; as a reference for $Ce^{4+}$, we measured the spectrum of $CeO_2$ in $O_2$ at 300 °C. The reference spectra were fitted using the peak positions found in a previous study[43]. The parameters (BE, FWHM, G/L ratio, $v^0/v^I$ ratio and $v/v^{II}/v^{III}$ ratio) were then applied to the peak fitting of all the *Ce 3d* spectra. Supplementary Table 9 lists the parameters for the *Ce 3d* fitting and Supplementary Fig. 21 gives the fitting details for all the *Ce 3d* spectra in this work. The measurement error bars associated with the area of different components obtained from the deconvolution of the spectra is the standard deviation calculated by repeating the fitting on a statistically relevant sample of spectra collected under the same reaction conditions.

## Density functional theory (DFT) calculations

DFT calculations were performed with the Quickstep code[75] within the CP2K package, using a mixed Gaussian and plane waves basis set, the Goedecker, Teter, and Hutter (GTH) pseudopotentials[76], and a GGA-PBE-D3[77] exchange-correlation functional[78]. Pseudopotentials with 30, 6, 4, and 18 valence electrons are used for Ce, O, C and Pt, respectively. We used a plane-wave basis-set energy cutoff of 1000 Ry and the Γ point to sample the Brillouin zone. The DFT + U method was used to describe the Ce 4f electrons. A U value of 4.5 eV was taken to obtain a balanced description of the different properties of ceria, in line with previous work with typical U values in the range of 3.0 to 5.5 eV[79,80]. The projected density of states (PDOS) is plotted using Gaussian smearing with a broadening parameter of 0.04 eV.

To study the electronic behavior of platinum nanoparticles (Pt NPs) around the critical size range identified, we created three (Pt NPs of sizes 0.9 nm (Pt0.9), 1.2 nm (Pt1.2), and 1.6 nm (Pt1.6), supported on the $CeO_2$ (111) surface. The number of Pt atoms are 25, 50 and 86 respectively. The p($5 \times 5$) $CeO_2$ (111) slab was chosen to support Pt0.9 and Pt1.2 NPs, and a p($6 \times 6$) $CeO_2$ (111) slab was chosen to support Pt1.6 NP, which is sufficiently large to isolate the NPs and prevent them from interacting with their periodic images. The $CeO_2$ (111) slabs have

three O-Ce-O tri-layers and the bottom tri-layer was fixed in the positions derived from the optimized bulk geometry.

To simulate $Ce^{3+}$ cations and the charge transfer process, similarly to ref. [63], we use a special norm-conserving 4f-in-core pseudopotential for the $Ce^{3+}$ cations[81]; this practically enforces a maximum oxidation state of 3+ for the atoms with this 4f-in-core pseudopotential (since the 4f electron is in the core and cannot be removed). For each nanoparticle, we first optimize the geometries with normal Ce pseudopotentials. Then, we substitute one Ce atom on the top layer with the 4f-in-core pseudopotential and optimize the structure again. Finally, we can calculate the relative stability of this structure with respect to the structure with normal pseudopotential by:

$$\Delta E = (E_{slab\_4fpp} + E_{Ce2O3\_bulk}) - (E_{slab} + E_{Ce2O3\_bulk\_4fpp}) \quad (1.1)$$

Where $E_{slab}$ is the energy of the surface slab with Pt nanoparticle, and $E_{Ce2O3\_bulk}$ is the energy of bulk $Ce_2O_3$, and 4fpp means that one Ce atom is substituted by the 4f-in-core pseudopotential.

Based on our previous calculations[47,51], the $CeO_2$ (223) stepped surface has high stability and provides surface sites that can easily trap single-atom platinum species. Therefore, this surface and the step sites on it were considered for the calculation of single-atom platinum species. The sodium atom is placed on the step site and, in order to keep the whole system neutral, an OH group is attached to the sodium atom to form one NaOH species. The single-atom platinum species $PtOx$ (x = 0, 1, 2) are then placed on step sites neighboring the NaOH species. The positions of the two bottom layers of the $CeO_2$ surface are fixed. Optimizations are considered complete when atomic forces reach 0.02 eV/Å.

In this study, the adsorption energy was calculated as follows:

$$E_{ads} = E_{slab + PtOx} - E_{slab} - E_{Pt} - x/2E_{O2} \quad (1.2)$$

where $E_{slab+ptox}$, $E_{slab}$, $E_{Pt}$ and $EO_2$ are the calculated electronic energies of adsorbed species on the surface, a $CeO_2$ (223) surface with or without NaOH species, a bulk platinum and a gas-phase $O_2$ molecule, respectively.

For the XPS binding energy, all calculations were performed using the Quantum ESPRESSO package[80]. We used the PBE exchange-correlation functional[77], and the pseudopotentials from the SSSP PBE efficiency v1.2 library[82–85]. We used a Gaussian smearing of the occupation of the electronic states with a broadening parameter of 0.02 Ry. For structural optimizations, the k-point sampling of the Brillouin zone uses a uniform $7 \times 7 \times 7$ mesh for the surface and Γ point for the cluster. Kohn–Sham wavefunctions and charge density are expanded in plane waves up to a kinetic-energy cutoff of 60 and 800 Ry, respectively. Absolute binding energies of core electrons were calculated by employing a Δ-Kohn-Sham (ΔKS) scheme[86].

To address the effect of CO adsorption on the XPS binding energy for a small nanoparticle, we calculated the binding energies of a small Pt cluster with and without adsorbed CO. The 4f core level shift (CLS) of the corner site without CO adsorption is −0.7 eV with respect to the calculated BE for a platinum atom in the bulk. CO adsorption leads to a binding energy shift in an opposite direction by 0.7 eV. Therefore, the effect of CO adsorption on a corner site of a platinum NP results in a CLS of essentially 0 eV with respect to the bulk.

The reaction energy is calculated as the difference between the energy of the structure with the adsorbed *COOH species and that of the structure with the dissociated *CO₂ and *H species.

## In situ gas-cell STEM and STEM-EELS Experiments
In situ gas-cell STEM experiments were performed on the DENSsolutions Climate GVB system. Powder catalyst (NP_0 fresh) was loaded into a micro-electro-mechanical system (MEMS) based nanoreactor. The catalysts were initially recalcined in the air (500 °C, 1 bar) in the

microscope to avoid carbon contamination, then cooled to RT. Subsequently, reaction mixture (0.2 mbar $CO_2$ + 0.2 mbar $H_2$ + 200 mbar $N_2$, 0.05 mL min⁻¹) was introduced and the nanoreactor was ramped up to 500 °C then cooled down to 250 °C for data acquisition. The NP_0 fresh sample and a model $Pt/CeO_2$[87] was also measured in higher pressure reaction mixture ($CO_2$:$H_2$ = 1:3, 1 bar, 0.3 mL min⁻¹) at 300 °C Images were acquired at a frame time -0.8 s with the approximately dose rate 500 e⁻/Å²·s. The electron beam was blanked when images were not being acquired. The time-resolved images were temporally binned by summing 5 frames, producing time-averaged images with an increased signal-to-noise ratio.

The model $Pt/CeO_2$ catalysts used for EELS analysis in this study were prepared by sputtering Pt nanoparticles onto $CeO_2$ powder (Aldrich 544841). To achieve this, the $CeO_2$ powder was first dispersed in methanol and then ground using a mortar. The resulting mixture was drop cast onto a copper TEM-grid that was covered with a lacey carbon film. After allowing the methanol to evaporate at room temperature, the grid was loaded into a sputter coater to apply the Pt NPs. Subsequently, the sample underwent plasma cleaning with an argon-oxygen mixture to remove any residual contaminants before its introduction into the electron microscope.

The analysis of the sample was conducted at room temperature under vacuum using a probe-corrected Titan Themis microscope operating at 300 kV. The beam convergence angle was set to approximately 26 mrad, with the approximately dose rate $2.4 \times 10^4$ e⁻/Å²·s. EEL spectra were recorded using the same microscope, with an ELA hybrid pixel direct electron detector from Dectris that was retrofitted with a CEFID energy filter from CEOS. The energy dispersion of the ELA detector was 1.245 eV per channel, and the dwell time was set to 1 s. Representative raw spectra are provided in Supplementary Fig. 35. Prior to analysis, the background of the EEL spectra was removed using Plural Scattering correction, a decaying power-law function to the pre-edge region and a double arctan background[88] at higher energy loss (Supplementary Fig. 35C). The valence state of $CeO_2$ was determined by calculating the ratio of the peak areas of $M_5$ to $M_4$. The relative thickness of $CeO_2$ was determined by calculating the ratio of zero-loss electrons to the total transmitted intensity[89] in units of the local inelastic mean free path λ using DigitalMicrograph software as listed in Supplementary Table 15.

## Ex situ X-ray diffraction (XRD)
The ex situ XRD measurements were carried out in the 2θ range of 20–65° on Bruker D8 advance with Cu $K_\alpha$ radiation (λ = 0.15406 nm) as the X-ray source. A thin layer of fine-ground catalyst powder was dispersed on a zero-background silicon wafer, placed in a sample holder and scanned with an increment step of 0.05° by keeping a time/step of 5 s.

## Data availability
The main data supporting the findings of this study are available within the paper and its Supplementary Information. Additional data are available from the corresponding authors upon request.

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

## Acknowledgments

This work received financial support from the SNF Grant 196946 (LA), SNSF project 200021_178943 (AB, JAvB), SNF Professorship Grant No. PP00P2_187185 (UA), SNSF project 200021_196381(XL, HE, RE), NCCR MARVEL, a National Center of Competence in Research, funded by the Swiss National Science Foundation (grant number 205602), and CSC Scholarship (XL, Qianyu Liu). We acknowledge the Swiss Light Source synchrotron for providing beamtime at the In Situ Spectroscopy

beamline (proposal ID 20212011). We thank Man Guo for support with the data analysis. We acknowledge Marcia Schoenberg for proofreading the manuscript. DFT calculations were performed on UBELIX, the HPC cluster at the University of Bern, and on Eiger from CSCS (production project mr32).

## Author contributions

X.L., L.A., and J.A.v.B. conceived the project. XL synthesized the catalysts and performed the catalytic tests. XL and Qiang Liu did the APXPS experiments. X.W. performed the DFT calculations. A.B., F.K., and X.L. performed the STEM and EDXS study. X.L. carried out the in situ STEM investigation. X.L. and H.E. performed the ex situ STEM-EELS study. M.A., Qianyu Liu, and X.L. contributed to the kinetic models. G.P., R.E., U.A., L.A., and J.A.v.B. supervised the study. X.L., L.A., and J.A.v.b. wrote the paper. All authors contributed to the discussion of the results and the manuscript revision.

## Competing interests

The authors declare no competing interests.
