## [Transparent Peer Review file · Nature Communications]

Quantifying electronic and geometric effects on the activity of platinum catalysts for water-gas shift

Corresponding Author: Dr Luca Artiglia

This manuscript has been previously reviewed at another journal. This document only contains information relating to versions considered at Nature Communications. Mentions of the other journal have been redacted.

Version 0:

Reviewer comments:

Reviewer #1

(Remarks to the Author)

The main result advocated in this work is that “the intrinsic activity of platinum atom at the perimeter corner sites is THREE ORDERS OF MAGNITUDE HIGHER AS A RESULT OF AN ELECTRONIC STRUCTURE EFFECT, WITH A THRESHOLD OCCURRING AT AN AVERAGE NANOPARTICLE SIZE OF 1-1.5 NM.” Such a phenomenon of quite abrupt electronic structure change of the corner Pt atoms in a narrow size range of ceria-supported nanoparticles, if convincingly verified, would be important not only for catalysis, but also for nanoscience in general.

In the present study the conclusion on the mainly electronic structure origin of the experimentally observed strong WGS catalytic activity enhancement was to a significant extent derived from DFT calculations. The calculated Bader charges of the corner Pt atoms (Fig. S25) change from ca. -0.04 au for the 0.9 nm large particle (below the threshold range of 1-1.5 nm) to ca. -0.08 au for both the 1.2 nm and 1.6 nm large particles (within the threshold range and above the threshold range, respectively). This quite small variation of charges seems incompatible with the expected change of the interaction energies of reactants and activation energies on these sites by 0.2 to 0.3 eV due to solely the charge change. In fact, the variations of CO adsorption energies on the unrelaxed particles, -2.2 eV (Pt0.9), -2.1 eV (Pt1.2) and -2.28 eV (Pt1.6) in Table R1-2, quantify the electronic structure effect on the reactivity to be not more than ca. 0.1 eV. Thus, some other effects appear to be important for the observed reactivity increase, beyond the mere electronic structure/charge effect. If the particle size effect on the activation energies of the WGS reaction is too difficult to calculate, one can opt for simpler calculations of the reaction energies, from which scaling relations would allow to evaluate, whether the activation energies lowering due to merely electronic structure effect is compatible with the observed reactivity increase by three orders of magnitude.

There are several other issues in the present manuscript, which also need to be also after clarifying the main issue outlined above. Just a couple of examples (the list could be extended):

a) Why some data are provided for the models where a Ce³⁺ cation (with 4f¹ electron configuration, i.e. with one more electron than in Ce⁴⁺ cation) was substituted by a Ce⁴⁺ cation? The number of Ce³⁺ cations induced by deposition of Pt particles on ceria is indeed an indicator of the charge transfer from Pt to the ceria support. But introducing Ce³⁺ cation and thus increasing the number of electrons in the system results in electron accumulation on Pt particles, opposite to the electron donation to the substrate. In addition, it should be better explained, why the present GGA+U calculations did not result in such electron donation, which is well documented in the literature.

b) The meaning of the Table S14 entitled “Relative stability of structure with one Ce³⁺ cation with respect to the structure with no Ce³⁺ for the three NP sizes considered here and for a pure CeO₂ surface without nanoparticle, and the charge transferred from Pt to the CeO₂ support, normalized by the number of Pt atoms in the 3D NP” is unclear as well as the reasons for publishing these data.

Reviewer #2

(Remarks to the Author)

Previous literature reports demonstrated the strong dependence of the observed WGS (water-gas-shift reaction) activity on the Pt particle size when they are supported on CeO₂ support: smaller Pt particles provide much higher WGS activity. Furthermore, it has been reported that the WGS activity strongly depends on the total number of Pt perimeter atoms which are in intimate contact with the CeO₂ support surfaces—leading to strong interactions.

In this work, the authors tried to further identify the most active site(s) in the Pt/CeO₂ powder catalyst system. They strongly relied on deconvoluting operando XPS data based on models which were derived from scanning transmission electron microscopy (STEM) images. Through extensive analysis (parameter fitting of XPS spectra), the authors concluded “the intrinsic activity of platinum atom at the perimeter corner sites is three orders of magnitude higher as a result of an electronic structure effect, with a threshold occurring at an average nanoparticle size of 1-1.5 nm.” While such a conclusion is intriguing more convincing experimental evidence needs to be provided. For this reason, this reviewer does not support the publication of this manuscript.

Listed below are a few questions/comments for the authors to consider. There are many other technical questions. The reviewer, however, considers these as critical.

1) With particle sizes of ~ 1 nm, would the concept of “corner”, “edge”, or “terrace” atoms be still a valid description for the nature of the metal species? In this size range, their “shape” or configuration should strongly depend on the nature of the support surfaces. It is expected that the strong metal-support interactions dominate the metal configurations and modify their electronic structures as well. Both the heterogeneities on the support surfaces and the different sizes of the metal particles contribute to the “broadening” of the experimentally observed XPS spectra. From this perspective, one could not simply classify the active sites as “corner”, “edge” and “terrace” sites on the metal particles. In fact, the so called “corner” sites should not be considered catalytically equivalent since the catalytic property of each individual “corner” site depends on the size of the metal particle and how that specific particle interacts with the support surface.

2) The authors should provide details on the validity of using XPS spectra to determine the corner, edge, and terrace sites with respect to the particle sizes. Does this method work for any sizes of the Pt particles? Which Pt sizes are most appropriate for using such a deconvolution method?

3) At the bottom of page 6, the authors stated “The sintering of AD platinum into NPs activates the catalyst towards the low-temperature WGS reaction (Figure 1(F)). The majority of platinum remains dispersed (as AD species) and it probably contributes little to the catalytic activity.” This is very confusing and misleading. The STEM image (Figure 1C) clearly shows that the AD species sintered to form Pt particles. There should not exist many AD species at all after the WGS. Even if the majority of the Pt species were AD species, then these species might have contributed significantly to the observed activity since these AD species were now reduced during the WGS process.

4) Figure 1 caption: “(D) Experimental XP spectra and peak fitting of the Pt 4f region obtained in 0.1 mbar CO + 0.3 mbar H₂O, during ramp-up at 100–300 °C and cool-down at 300–100 °C, using 620 eV of photon energy. Pt 4f region contains four components corresponding to bulk atoms (a), terrace atoms with bound CO (b), low-coordinated atoms with bound CO (c) and atomically dispersed (AD) Pt²⁺ ions (d).” If the authors’ interpretation of the XPS spectra were correct, then most of the AD species did not sinter during the WGS process, contradicting the STEM imaging results and those in literature reports. If the most dominant peak in the XPS spectra (Figure 1d) does not originate from the AD species, then the rest of the analyses of the XPS spectra in this manuscript becomes irrelevant.

5) Page 10, line 13. “This is because XPS aids in accurately quantifying AD and NP species, thereby complementing the structural insights obtained from STEM.” This statement is not correct or at least misleading. There are no validated reports to show that XPS technique can be used to quantify the amount of AD species in a supported metal catalyst.

6) In many places (main text and supporting information), it is not clear if the authors refer to general “corner” sites of a Pt nanoparticle or the specific “perimetric corner” sites of a supported Pt nanoparticle. These two different sites should be catalytically very different. If the authors refer to the “perimetric corner” sites, then some of the conclusions are not necessarily correct. For example, the number and nature of “perimetric corner” sites should be very different for “pancake-like” and “diamond-like” Pt nanoparticles. In fact, for smaller Pt particles, they usually do not show any clearly identifiable segments of perimeters. Therefore, the definition of “perimetric corner” sites becomes ambiguous or meaningless.

Version 1:

Reviewer comments:

Reviewer #3

(Remarks to the Author)

The authors have made a significant advancement in understanding water-gas shift (WGS) activity by integrating APXPS signal deconvolution results with STEM analyses to quantify the proportion of different Pt sites in the catalyst under varying conditions. This work establishes a quantitative relationship between WGS activity and specific Pt sites, advancing the widely accepted understanding of performance by pinpointing the critical role of corner atoms within the perimeter atoms, rather than just the general perimeter atoms. This progress, achieved through considerable time and effort, is highly commendable. However, rigorous language expression and analytical logic are essential in research to prevent catastrophic issues, and thus, the article requires further optimization in these points.

1. On page 6, line 5, the article states, “Therefore, platinum in our samples is subject to the laws of thermodynamic stability, leading to the formation of NPs as the predominant species.” This contradicts another statement: “The majority of platinum

remains dispersed (as AD species)." Such contradictions, along with other instances of unclear expression, appear multiple times throughout the article, undermining its clarity and reliability.

2. In exploring the structure-performance relationship, the authors establish a linear correlation between small particle size and performance, focusing the study on this size range. Subsequently, a linear relationship between corner atoms on small particles and performance is identified. However, it is one-sided to assert, based on this analysis, that "This finding challenges the assumption that the corner sites are equally active regardless of NP size, and that the mere geometric effect is responsible for the size effect of supported platinum for WGS".

Reviewer #4

(Remarks to the Author)

Comments to NCOMMS_504207: Quantifying Electronic Structure and Geometric Effects on the Activity of Pt/CeO₂ Catalysts for the Water-Gas Shift Reaction

This manuscript presents a comprehensive study of the structure–activity relationship in Pt/CeO₂ catalysts for the water-gas shift (WGS) reaction, combining operando X-ray photoelectron spectroscopy (XPS), scanning transmission electron microscopy (STEM), electron energy-loss spectroscopy (EELS), density functional theory (DFT) calculations, and kinetic modeling. The authors report that Pt atoms located at perimeter corner sites exhibit an intrinsic activity three orders of magnitude higher than other sites, primarily due to electronic structure effects, with a threshold particle size identified at approximately 1–1.5 nm. The manuscript is well-written, and the rebuttal letter thoroughly addresses the previous reviewers' comments.

However, I have several concerns regarding the robustness of the conclusions, particularly due to the heavy reliance on XPS fitting and the extrapolation of STEM data. These concerns warrant a major revision before the manuscript can be considered for publication.

Major comments:

1. XPS Fitting and Site Deconvolution Accuracy:

The conclusion relies heavily on deconvoluting the Pt 4f spectra to assign site-specific signals (bulk, terrace, edge, corner, AD). However, the binding energy differences between these components—particularly edge, corner1, and corner2—are small (often <0.5 eV), and fall within the resolution limits of XPS. This overlap raises concerns about the reliability and reproducibility of site-specific quantification, especially when these subtle differences drive the key conclusions.

2. Use of STEM to Guide Global XPS Interpretation:

The authors use localized STEM images to inform the morphology and shape of Pt nanoparticles, which is then used to constrain XPS fitting across the entire catalyst. However, XPS is a global technique, averaging over a much larger sample area than STEM. The limited number of STEM images cannot reliably represent the full sample, raising questions about whether the assumed NP geometries and distributions are truly reflective of the bulk material.

3. STEM Conditions vs. Operando Environment:

Moreover, the STEM data were acquired under high vacuum, whereas the XPS data were collected under WGS reaction conditions (gas flow and elevated temperature). Pt NP morphology and surface structure can change significantly under gas environments. While the authors cite prior in situ TEM studies and present ex situ STEM images, the referenced in situ work used different gas compositions and temperatures. These conditions may not accurately reflect the current system. The structure of Pt NPs was reported to go through very dynamic changes under reaction conditions, such as in the following recent papers: <https://www.nature.com/articles/s41467-021-26047-8>, <https://www.science.org/doi/10.1126/science.ads2688>, <https://www.nature.com/articles/s43246-024-00575-4#Sec13>. For greater confidence in the particle shape assignment under relevant conditions, in situ TEM data under the same WGS conditions used in this study would be highly valuable.

Minor Points:

1. Figure 1 – AD vs NP Claim:

The claim that only AD Pt is present before WGS, and only aggregated NPs are observed afterward, is not fully supported by the current STEM images in Fig. 1 and Fig. S5. The authors should provide images at same magnification at both atomic scale and lower magnification to show representative regions of the CeO₂ support before and after reaction.

2. EELS – Beam-Induced Reduction of CeO₂:

In Fig. S31, EELS spectra are only shown for regions near the Pt/CeO₂ interface. Since electron beams can reduce Ce⁴⁺ to Ce³⁺, a reference EELS spectrum from bulk CeO₂ under identical acquisition conditions is needed to confirm that the observed changes are due to Pt–support interaction rather than beam effects.

3. EDX Data Quality – Fig. S36:

The EDX spectra shown have low counts, making it unclear whether the absence of Na peaks reflects true absence or insufficient acquisition time. The authors should improve the EDX signal quality or clarify the capture parameters used.

4. Gas Pressure Clarification – Page 4, Lines 93–94 and Fig. 1A:

The feeding composition is given as 100 ppm CO and 300 ppm H₂O, but the total pressure is not specified. Please clarify the actual pressure used in these flow reactor experiments.

5. Microscope Specification – Methods Section:

Instead of broadly stating "aberration-corrected TEM," please specify whether the instrument used was probe-corrected or image-corrected to clarify imaging capabilities. In S1.6, Line 1129, "the spot size was adjusted to keep the beam current low in order to minimize beam-effects", please specify the spot size and dose rate.

Recommendation:

I recommend major revision to address the concerns above. While the manuscript presents a novel and thorough investigation into the electronic structure effects of Pt/CeO₂ catalysts, the key conclusions rest on assumptions that require

more robust experimental validation. Addressing the issues related to XPS–STEM correlation and in situ structural fidelity will significantly enhance the strength and credibility of the work.

Version 2:

Reviewer comments:

Reviewer #3

(Remarks to the Author)

The authors have revised the manuscript according to the comments and suggestions from Reviewers. It is recommended for publication in Nature Communications.

Reviewer #4

(Remarks to the Author)

In the revised manuscript, the authors have adequately addressed the concerns I raised in the initial review. The updates enhance the presentation and improve the manuscript's clarity and coherence. I commend the authors for their responsiveness. I recommend to accept this paper.

One minor point is about the in-situ STEM data, in the labeled time stamp '02:50', the time format should be clarified — is this in mm:ss or hh:mm? Specifying the format would enhance clarity for the reader.

Comments in *blue* - Replies in black - Actions in *green*.

Reviewer #1:

- The main result advocated in this work is that "the intrinsic activity of platinum atom at the perimeter corner sites is THREE ORDERS OF MAGNITUDE HIGHER AS A RESULT OF AN ELECTRONIC STRUCTURE EFFECT, WITH A THRESHOLD OCCURRING AT AN AVERAGE NANOPARTICLE SIZE OF 1-1.5 NM." Such a phenomenon of quite abrupt electronic structure change of the corner Pt atoms in a narrow size range of ceria-supported nanoparticles, if convincingly verified, would be important not only for catalysis, but also for nanoscience in general.

Thank you for recognizing the high and broad impact and importance of the main conclusion of our work. This work is the result of a PhD work (3 years). The experimental results have been reproduced several times, to validate their consistency. The group has access to an ambient pressure X-ray photoelectron spectroscopy (APXPS) beamline at the Swiss light source (in house research) and has a solid experience in data acquisition, data treatment and analysis. This manuscript presents a unique approach to investigate a real (powder) catalyst combining operando spectroscopy with high resolution scanning electron microscopy/electron energy loss spectroscopy, reactivity data and density functional theory calculations. Therefore, it is not only showing important scientific results, but is also proposing a new method for the characterization of structure activity relationships.

- In the present study the conclusion on the mainly electronic structure origin of the experimentally observed strong WGS catalytic activity enhancement was to a significant extent derived from DFT calculations. The calculated Bader charges of the corner Pt atoms (Fig. S25) change from ca. -0.04 au for the 0.9 nm large particle (below the threshold range of 1-1.5 nm) to ca. -0.08 au for both the 1.2 nm and 1.6 nm large particles (within the threshold range and above the threshold range, respectively). This quite small variation of charges seems incompatible with the expected change of the interaction energies of reactants and activation energies on these sites by 0.2 to 0.3 eV due to solely the charge change. In fact, the variations of CO adsorption energies on the unrelaxed particles, -2.2 eV (Pt0.9), -2.1 eV (Pt1.2) and -2.28 eV (Pt1.6) in Table R1-2, quantify the electronic structure effect on the reactivity to be not more than ca. 0.1 eV. Thus, some other effects appear to be important for the observed reactivity increase, beyond the mere electronic structure/charge effect. If the particle size effect on the

activation energies of the WGS reaction is too difficult to calculate, one can opt for simpler calculations of the reaction energies, from which scaling relations would allow to evaluate, whether the activation energies lowering due to merely electronic structure effect is compatible with the observed reactivity increase by three orders of magnitude.

We appreciate the suggestion to further strengthen the robustness of this work. DFT calculations have been used to *support* experimental data (e.g., providing NPs structure simulations), and mainly as requested by reviewers in the previous review round in [redacted]. We spent considerable efforts and resources trying to address all the comments.

We conducted additional calculations to evaluate the reaction energies as suggested. This is not trivial because several reaction mechanisms have been proposed for WGS (see below). The reaction energy was determined as the difference between the energy of the structure with the adsorbed *COOH species and that of the structure with the dissociation products *CO₂ and *H species, the rate-determining step (RDS) for the associative carboxyl mechanism in the water-gas shift reaction on Pt/CeO₂. As shown in Figure R1-1, the smaller platinum nanoparticles on CeO₂ (Pt0.9) exhibits lower reaction energies than the larger ones (Pt1.6). The approximately 0.3 eV energy difference, consistently observed from both the median and mean values, provides strong evidence for the differences in the intrinsic activity of corner atoms on the small respectively large Pt NPs.

Figure R1-1. Distribution of reaction energies for the carboxyl dissociation on 0.9 nm (Pt0.9) and 1.6 nm (Pt1.6) Pt NPs on CeO₂.

Below, we provide details of the computational model and results. Various reaction pathways, including the "**redox**" pathway (*J. Catal.* 2011, 279 (2), 287-300; *Catal. Today* 2007, 126 (1), 143-147; *Applied Catalysis B: Environmental* 2013, 136-137, 225-238) and the "**associative**" pathway (*J. Phys. Chem. C* 2011, 115, 23, 11595–11610; *J. Phys. Chem. C* 2014, 118 (12), 6314-6323; *Applied Catalysis A: General* 2004, 268 (1), 255-266), have been proposed for WGS mechanism over Pt/CeO₂. Our previous experimental findings revealed an associative reaction mechanism via carboxyl pathway on the same Pt/CeO₂ samples examined in the present study (*ETH Zurich Doctoral Thesis*, Chapter 6, doi.org/10.3929/ethz-b-000620557). The RDS is the dissociation of carboxyl (*COOH), which is in line with the theoretical calculations (*J. Phys. Chem. C* 2014, 118, 12, 6314–6323; *J. Phys. Chem. C* 2008, 112, 12, 4608–4617). Therefore, the carboxyl pathway was selected for our modeling and we exclusively focused on the dissociation of carboxyl step in our analysis for the evaluation of activation energies.

Figure R1-2 illustrates the atomic structures for the small (0.9 nm, naming Pt0.9) and large (1.6 nm, naming Pt1.6) Pt NP models on CeO₂ (111) slab, modeled by a p(6 × 6) unit cell and three atomic layers. Both models exhibit a truncated octahedral shape with four highlighted corners. The NPs were sampled across various (eight) positions and orientations on the surface, and the most stable geometries were selected for the reaction. Given the strong adsorption of CO on platinum particles, we assumed full CO coverage on Pt surface atoms that do not directly interact with the CeO₂ substrate. The surface chemistry of CeO₂ is inherently complex due to the presence of oxygen vacancies and possible hydroxyl (OH) group coverage. However, since our focus is on the *COOH dissociation process occurring on Pt NPs, we employed a clean CeO₂ surface to prevent the introduction of additional complexities unrelated to this reaction step.

Figure R1-2. The side and top view of the atomic structures of (A) Pt0.9, and (B) Pt1.6, supported on CeO₂(111). The colors of the Ce, O, Pt and C atoms are yellow, red, light-grey, and dark-grey respectively. Corner Pt atoms are highlighted with green spheres.

Figure R1-3 illustrates the atomic structures and reaction energies during *COOH dissociation on 0.9 nm and 1.6 nm Pt nanoparticles. To ensure statistical validity and representativeness, we analyzed one Pt site from each of the four corner locations, resulting in a total of four Pt sites. Reaction energies vary depending on the local environment of the active Pt sites. In Panel B4, we observed an unusually high reaction energy of 1.1 eV, attributed to the direct desorption of CO₂ following the dissociation of *COOH. Unlike other cases where CO₂ remains adsorbed on Pt, this energy is not directly comparable to other values and has not been included in the subsequent analysis. To ensure completeness and transparency, we have also provided an analysis including the 1.1 eV value in the Supporting Information, where the statistical significance remains. The mean and median reaction energies are 0.20 eV and 0.195 eV, respectively, for Pt0.9, compared to 0.52 eV and 0.44 eV for Pt1.6. Although activation energies cannot be directly averaged due to their impact on the reaction rate being via an exponential function, the distribution of reaction energies shown in Figure R1-1 reveals a statistically significant difference between the two groups, with a p-value of 0.029 (one-tailed Mann-Whitney U test). The approximately 0.25-0.3 eV

reaction energy difference in the RDS reflects a substantial difference in activation energies of WGS, based on the scaling relations (*J. Am. Chem. Soc.* 2003, 125, 13, 3704–3705; *Phys. Chem. Chem. Phys.*, 2005, 7, 2552–2553). The reaction order of CO₂ was determined to be zero in the experimental kinetic tests, while H₂ exhibits a negative reaction order in both our kinetic results in Figure S34 and previous literature (e.g., *Catalysis Today* 123, 1–4, 2007, 224–234). This suggests that the observed differences in the dissociation reaction energies are primarily associated with hydrogen. To probe this further, we examined the *H adsorption energies. Specifically, after obtaining the optimized geometry for the *COOH dissociation pathway, we removed the adsorbed CO₂ and re-optimized the structure to isolate the influence of *H adsorption. The results indicate a statistically significant difference, with *H adsorption energy being notably higher on Pt0.9 compared to Pt1.6, as highlighted in Figure R1-4.

Figure R1-3. The atomic structures of the *COOH dissociation process on (A) Pt0.9 and (B) Pt1.6. Each panel displays key intermediates of the reaction step at the bottom. To maintain the overall stoichiometry of the WGS, a hydrogen atom is placed on a neighboring lattice oxygen site. The lattice cerium site, where OH was initially adsorbed before forming COOH on the Pt site, is also shown to indicate its origin. In the structure model, only the relevant atoms are highlighted as spheres, and the rest are represented as lines. The colors of the Ce, O, Pt, C and H atoms are yellow, red, light-grey, dark-grey, and white, respectively.

Figure R1-4. Distribution of *H adsorption energy after carboxyl dissociation on 0.9 nm (Pt0.9) and 1.6 nm (Pt1.6) Pt NPs on CeO₂.

We have added the above discussions on Page 14, S1.5, S2.7, and Figures S25-28 to further support our claims.

Page 14

The substantial difference in the intrinsic activity between larger and smaller NPs can be corroborated by theoretical calculations of the reaction energy for the rate-determining step (RDS), which is typically considered to be the dissociation of the carboxyl in the associative mechanism of WGS (Grabow, Gokhale et al. 2008, Aranifard, Ammal et al. 2014). As discussed further in Sections S1.5 and S2.7 and illustrated in Figures S25–28, the corner atoms on the 0.9 nm and 1.6 nm Pt NP models exhibit a difference of approximately 0.3 eV in reaction energy, demonstrating substantial difference in the activation barrier from their scaling relation (Michaelides, Liu et al. 2003, Inderwildi, Lebedez et al. 2005). This provides strong evidence for the pronounced disparity in the intrinsic activity of NPs of varied size.

S1.5

The reaction energy is calculated as the difference between the energy of the structure with the adsorbed *COOH species and that of the structure with the dissociated *CO₂ and *H species.

S2.7 Reaction energy analysis

Reaction energy calculations have been performed on two Pt NPs at sizes of 0.9 nm (Pt0.9) and 1.6 nm (Pt1.6), supported on the CeO₂ (111) surface. The number of Pt

atoms are 25 and 86 respectively. Figure S25 illustrates the atomic structures for the small (0.9 nm, naming Pt0.9) and large (1.6 nm, naming Pt1.6) Pt NP models on CeO₂ (111) slab, modeled by a p(6 × 6) unit cell and three atomic layers. Both NPs exhibit a truncated octahedral shape with four highlighted corners. The NPs were sampled across various (eight) positions and orientations on the surface, and the most stable geometries were selected for the reaction. Given the strong adsorption of CO on platinum particles, we assumed full CO coverage on Pt surface atoms that do not directly interact with the CeO₂ substrate. The surface chemistry of CeO₂ is inherently complex due to the presence of oxygen vacancies and possible hydroxyl (OH) group coverage. However, since our focus is on the *COOH dissociation process (as discussed in the following paragraph) we employed a clean CeO₂ surface to prevent the introduction of additional complexities unrelated to this reaction step.

Various reaction pathways, including the "redox" pathway ((Meunier, Tibiletti et al. 2007, Kalamaras, Americanou et al. 2011, Kalamaras, Petalidou et al. 2013)) and the "associative" pathway ((Jacobs, Khalid et al. 2004, Kalamaras, Gonzalez et al. 2011, Aranifard, Ammal et al. 2014)), have been proposed for WGS mechanism. Our previous experimental findings revealed an associative reaction mechanism via carboxyl pathway on the same Pt/CeO₂ samples examined in the present study ((Li 2023)). The rate-determining step is the dissociation of carboxyls (*COOH), which is in line with other theoretical calculations ((Grabow, Gokhale et al. 2008, Aranifard, Ammal et al. 2014)). Therefore, the carboxyl pathway was selected for our modeling, and we exclusively focused on this step in our analysis for the evaluation of activation energies based on the scaling relation between reaction energy and activation energy. The carboxyl pathway can be described by the following intermediate steps (IMs):

IM9: $^*_{\text{Pt}} - ^*_{\text{(O)}} - ^*_{\text{Ce}}$

$^*\text{COOH}$ dissociation process (IM5→IM6) is the rate-limiting step of the carboxyl pathway, thus we focused calculations on this step.

Figure S26 illustrates the atomic structures and reaction energies during $^*\text{COOH}$ dissociation on 0.9 nm and 1.6 nm Pt nanoparticles. To ensure statistical validity and representativeness, we analyzed one Pt site from each of the four corner locations, resulting in a total of four Pt sites. Reaction energies vary depending on the local environment of the active Pt sites. In Panel B4, we observed an unusually high reaction energy of 1.1 eV, attributed to the direct desorption of CO_2 following the dissociation of $^*\text{COOH}$. Unlike other cases where CO_2 remains adsorbed on Pt, this energy is not directly comparable to other values and has not been included in the subsequent analysis. The mean and median reaction energies are 0.20 eV and 0.195 eV, respectively, for Pt0.9, compared to 0.52 eV and 0.44 eV for Pt1.6. Although activation energies cannot be directly averaged because their impact on the reaction rate is via an exponential function, the distribution of reaction energies shown in Figure S27(A) reveals a statistically significant difference between the two groups, with a p-value of 0.029 (one-tailed Mann-Whitney U test). The approximately 0.25-0.3 eV reaction energy difference in the RDS reflects a substantial difference in activation energies, based on the scaling relations. This aligns with the pronounced disparity in intrinsic activities between small and large Pt nanoparticles. The lower activation barrier for the 0.9 nm Pt nanoparticle is likely related to the adsorption energy of the reaction products. Since the reaction order of CO_2 was determined to be zero in the kinetic tests, we focused on the adsorption behavior of $^*\text{H}$ species, as H_2 exhibits a negative reaction order in both our kinetic results in Figure S34 and previous literature (Phatak, Koryabkina et al. 2007). For $^*\text{H}$ adsorption energy calculations, after obtaining the optimized geometry for the $^*\text{COOH}$ dissociation pathway, we removed the adsorbed CO_2 and re-optimized the structure to isolate the influence of $^*\text{H}$ adsorption. As highlighted in Figure S27(B), the $^*\text{H}$ adsorption energy on Pt0.9 remains statistically significantly higher than that on Pt1.6, which is fully consistent with the observed difference in their reaction energies. To ensure completeness and transparency, we have also provided an analysis including the 1.1 eV value in Figure S28, where the statistical significance remains.

Figure S25. The side and top view of the atomic structures of (A) Pt0.9, and (B) Pt1.6, supported on CeO₂(111). The colors of the Ce, O, Pt and C atoms are yellow, red, light-grey, and dark-grey respectively. Corner Pt atoms are highlighted with green spheres.

Figure S26. The atomic structures of the *COOH dissociation process on (A) Pt0.9 and (B) Pt1.6. Each panel displays key intermediates of the reaction step at the bottom. To maintain

the overall stoichiometry of the WGS, a hydrogen atom is placed on a neighboring lattice oxygen site. The lattice cerium site, where OH was initially adsorbed before forming COOH on the Pt site, is also shown to indicate its origin. In the structure model, only the relevant atoms are highlighted as spheres, and the rest are represented as lines. The colors of the Ce, O, Pt, C and H atoms are yellow, red, light-grey, dark-grey, and white, respectively.

Figure S27. Distribution of reaction energies for the carboxyl dissociation and *H adsorption energy after carboxyl dissociation on 0.9 nm (Pt0.9) and 1.6 nm (Pt1.6) Pt NPs on CeO₂.

Figure S28. Distribution of reaction energies for the carboxyl dissociation on 0.9 nm (Pt0.9) and 1.6 nm (Pt1.6) Pt NPs on CeO₂ when taking the 1.1 eV reaction energy into account.

There are several other issues in the present manuscript, which also need to be also after clarifying the main issue outlined above. Just a couple of examples (the list could be extended):

a) Why some data are provided for the models where a Ce³⁺ cation (with 4f¹ electron configuration, i.e. with one more electron than in Ce⁴⁺ cation) was substituted by a Ce⁴⁺ cation? The number of Ce³⁺ cations induced by deposition of Pt particles on ceria is indeed an indicator of the charge transfer from Pt to the ceria support. But introducing Ce³⁺ cation and thus increasing the number of electrons in the system results in electron accumulation on Pt particles, opposite to the electron donation to the substrate. In addition, it should be better explained, why the present GGA+U calculations did not result in such electron donation, which is well documented in the literature.

Thank you for your thoughtful question. However, there seems to be some misinterpretation regarding the electron transfer mechanism in our models. We are not substituting Ce³⁺ with Ce⁴⁺; rather, we replace a Ce atom described by a normal pseudopotential with one using a 4f-in-core pseudopotential. This does not alter the total charge of the system but instead "freezes" a 4f electron within the core, effectively constraining the maximum oxidation state of that particular Ce atom. This approach is grounded in prior studies (e.g., *J. Catal.* 344, 507–514, 2016) showing that such pseudopotentials effectively capture Ce³⁺ behavior and electron transfer phenomena in Pt/CeO₂ systems. In our simulations of Pt/CeO₂ containing one Ce³⁺ ion, Bader charge analyses reveal that Pt atoms carry a net positive charge (around +0.3e⁻). Concurrently, the Bader charge on the Ce³⁺ species (~2.07e⁻) differs from that of Ce⁴⁺ in bulk CeO₂ (~2.4e⁻), signifying its reduced state. These observations collectively support the view that electrons flow from Pt to ceria.

Describing localized 4f states in Ce³⁺ accurately poses a well-known challenge in DFT due to self-interaction errors. Although the DFT+U method is widely used to address these issues, the result often depends strongly on the choice of the U parameter, which can vary among different studies and computational frameworks. A higher U value can indeed stabilize Ce³⁺, but it may distort other material properties (e.g., reaction energies, lattice parameters) away from experimental benchmarks.

In our previous work (*Phys. Chem. Chem. Phys.* 22, 28–38, 2020), we found that U = 4.5–5.0 eV provides a balanced description of properties such as lattice constants, band gaps, and redox energetics for CeO₂. Our study of single Pt atoms and Ce³⁺ on various CeO₂ surfaces showed that U = 5 eV allowed electron donation from Pt only on the stepped (112) surface, while U = 8 eV stabilized Ce³⁺ more broadly but significantly altered material properties. For instance, the CeO₂ to Ce₂O₃ reduction energy dropped from 355 kJ/mol (U = 5 eV) to 259 kJ/mol (U = 8 eV), deviating from the experimental 388 kJ/mol. To avoid excessively large U values that compromise other physical

properties, we adopt an alternative strategy: 4f-in-core pseudopotentials. By treating the 4f electron as part of the ionic core, one sidesteps many of the complications of partially filled f orbitals while still capturing the essential physics of Ce³⁺ sites. This method has been validated across multiple DFT codes (both plane-wave and Gaussian-type basis implementations like CP2K used in this work), ensuring robust descriptions of catalytic and redox phenomena without overly large U corrections.

We have modified the text in S2.8 to clarify this explanation.

Describing localized 4f states in Ce³⁺ accurately poses a well-known challenge in DFT due to self-interaction errors. Although the DFT+U method is widely used to address these issues, the result often depends strongly on the choice of the U parameter, which can vary among different studies and computational frameworks. A higher U value can indeed stabilize Ce³⁺, but it may distort other material properties (e.g., reaction energies, lattice parameters) away from experimental benchmarks.

In our previous work (Wang et al. 2020), we found that U = 4.5–5.0 eV provides a balanced description of properties such as lattice constants, band gaps, and redox energetics for CeO₂. Our study of single Pt atoms and Ce³⁺ on various CeO₂ surfaces showed that U = 5 eV allowed electron donation from Pt only on the stepped (112) surface, while U = 8 eV stabilized Ce³⁺ more broadly but significantly altered material properties. For instance, the CeO₂ to Ce₂O₃ reduction energy dropped from 355 kJ/mol (U = 5 eV) to 259 kJ/mol (U = 8 eV), deviating from the experimental 388 kJ/mol. To avoid excessively large U values that compromise other physical properties, we adopt an alternative strategy: 4f-in-core pseudopotentials. By treating the 4f electron as part of the ionic core, one sidesteps many of the complications of partially filled f orbitals while still capturing the essential physics of Ce³⁺ sites. This method has been validated across multiple DFT codes (both plane-wave and Gaussian-type basis implementations like CP2K used in this work), ensuring robust descriptions of catalytic and redox phenomena without overly large U corrections.

b) The meaning of the Table S14 entitled "Relative stability of structure with one Ce³⁺ cation with respect to the structure with no Ce³⁺ for the three NP sizes considered here and for a pure CeO₂ surface without nanoparticle, and the charge transferred from Pt to the CeO₂ support, normalized by the number of Pt atoms in the 3D NP" is unclear as well as the reasons for publishing these data.

Thank you for this comment. Table S14 shows the relative stability of the Ce^{3+} cation on the CeO_2 surface with and without Pt NPs. The relative stability (ΔE) is calculated by the following equation (20) in S2.8:

$$\Delta E = (E_{\text{slab_4fpp}} + E_{\text{Ce2O3_bulk}}) - (E_{\text{slab}} + E_{\text{Ce2O3_bulk_4fpp}}) \quad (20)$$

Here:

- E_{slab} is the energy of the surface slab using a normal Ce pseudopotential (all Ce in Ce^{4+}).
- $E_{\text{slab_4fpp}}$ is the total energy of the same slab when one Ce atom in the surface region is replaced by a 4f-in-core pseudopotential (thus forcing that particular Ce to become Ce^{3+}).
- $E_{\text{Ce2O3_bulk}}$ and $E_{\text{Ce2O3_bulk_4fpp}}$ are the energies of bulk Ce_2O_3 computed with the normal Ce pseudopotential and 4f-in-core pseudopotential, respectively.

Including the bulk Ce_2O_3 energies ensures that any systematic shift in the absolute energy arising from using two different pseudopotentials (normal vs. 4f-in-core) is canceled out, allowing the final ΔE value to reflect only the relative stability of creating a Ce^{3+} site on the surface.

The ΔE represents the energy required to generate a Ce^{3+} cation on the surface slab. The value decreased from 2.50 eV (for a pure CeO_2 surface) to a range of 0.37 to 0.47 eV (for a CeO_2 surface with a Pt NP on top). This indicates that Pt NPs play a significant role in stabilizing the surface with Ce^{3+} , thus facilitating the formation of Ce^{3+} .

We have modified the text in the S2.8 to enhance clarity.

By employing a 4f-in-core pseudopotential in the slab, we force the formation of a Ce^{3+} cation on the surface slab. Additionally, the energy of the bulk Ce_2O_3 is incorporated to cancel the absolute energy difference introduced by the two different pseudopotentials.

ΔE represents the energy required to generate a Ce^{3+} cation on the surface slab. We calculate the ΔE for slabs without Pt NP and slabs with different NPs. Table S14 shows the stability of the Ce^{3+} cation on the CeO_2 surface with and without Pt NPs. The values are positive in all cases, indicating that the Ce^{3+} cations are not stable in our model, which is consistent with the result of our DFT calculation with normal pseudopotentials, as discussed above. However, the value decreases from 2.50 eV (for a pure CeO_2 surface) to a range of 0.37 to 0.47 eV (for a CeO_2 surface with a Pt NP on top). This indicates that Pt NPs play a significant role to stabilize the surface, thus facilitating the formation of Ce^{3+} .

Reviewer #2:

Previous literature reports demonstrated the strong dependence of the observed WGS (water-gas-shift reaction) activity on the Pt particle size when they are supported on CeO₂ support: smaller Pt particles provide much higher WGS activity. Furthermore, it has been reported that the WGS activity strongly depends on the total number of Pt perimeter atoms which are in intimate contact with the CeO₂ support surfaces—leading to strong interactions.

Thank you for your summary of the literature on the water-gas shift reaction (WGS), which we also read and reported in the introduction of our study. We respectfully disagree with the assertion that "the WGS activity strongly depends on the total number of Pt perimeter atoms in intimate contact with the CeO₂ support surfaces." To our knowledge, no existing studies have quantitatively correlated the number and type of Pt-CeO₂ perimeter atoms with WGS activity. One of the closest relevant works is the study published in *Nat. Commun.* 12, 914, 2021, which employed in situ TEM to observe the dynamics of Pt atoms at the perimeter during WGS, in contrast to the relatively static Pt atoms at terrace/step surface sites. However, this study did not provide a **quantitative analysis** correlating the fraction of perimeter atoms to WGS activity. Ribeiro group correlated the number of **corner sites** on **Au/TiO₂** with WGS reaction rate, **independent of the nanoparticle size** under investigation (*J. Am. Chem. Soc.* 2010, 132, 40, 14018–14020; *J. Am. Chem. Soc.* 2012, 134, 10, 4700–4708; *J. Catal.* 293, 2012, 94-102). Additionally, Cargnello et al. (*Science* 341, 771-773, 2013) used electron microscopy to measure the size of monodisperse platinum nanocrystals and related the number of atoms at the Pt-CeO₂ interface (perimeter atoms) to **CO oxidation activity**. Similarly, Flytzani-Stephanopoulos et al. (*Adv. Funct. Mater.*, 18: 2801-2807) correlated the **CO oxidation** reaction rate with the total length of **Au/CeO₂** interfaces (perimeter atoms). While these studies highlight the significance of metal-support interfaces (perimeter atoms), they have not quantitatively linked the number of **Pt/CeO₂ perimeter atoms** to **WGS activity**. More importantly, our work does not merely reiterate the importance of perimeter atoms (as highlighted in previous studies); instead, we place a stronger emphasis on the critical role of perimeter atoms in **smaller** Pt nanoparticles, which can explain the counterintuitive observation where the further addition of platinum on 0.2 monolayer Pt on CeO₂(111) actually leads to a decrease in WGS activity (*J. Am. Chem. Soc.* 134, 8968-8974, 2012). Our study demonstrates that it is a combination of size (geometry) and electronic effects that gives Pt NPs their unique activity towards WGS.

In this work, the authors tried to further identify the most active site(s) in the Pt/CeO₂ powder catalyst system. They strongly relied on deconvoluting operando XPS data based on models which were derived from scanning transmission electron microscopy (STEM) images. Through extensive analysis (parameter fitting of XPS spectra), the authors concluded "the intrinsic activity of platinum atom at the perimeter corner sites is three orders of magnitude higher as a result of an electronic structure effect, with a threshold occurring at an average nanoparticle size of 1-1.5 nm." While such a conclusion is intriguing more convincing experimental evidence needs to be provided. For this reason, this reviewer does not support the publication of this manuscript.

We appreciate your feedback and concerns regarding the robustness of the experimental evidence supporting our conclusions. In response, we have provided new evidence on the reaction energy for nanoparticles below and above the critical size threshold.

As detailed in the response to Reviewer 1, we evaluated the reaction energies of the rate determine step in the water-gas shift reaction on Pt/CeO₂ for both small (0.9nm, Pt_{0.9}) and large (1.6nm, Pt_{1.6}) Pt models on CeO₂. The substantial energy difference, which is statistically significant, further verifies and rationalizes the pronounced disparity in their intrinsic activity.

Furthermore, we highlight the multi-faceted approach we employed to validate our findings, which includes the following key aspects:

-Kinetic data analysis: We fitted the WGS activity data using an Arrhenius kinetic model, achieving a high correlation coefficient ($R^2 = 0.95$). This model is not a simple proportional correlation between WGS activity and the number of CPS; it considers the overall activation energies and reaction rates at all measured temperatures, which strongly supports our conclusion regarding the activity of Pt perimeter atoms.

-STEM-EELS experiments: Our STEM-EELS experiments demonstrated a significant change in Pt-CeO₂ interaction at the interface as a function of Pt size, particularly at ~1.5 nm. This substantial increase in electronic interaction corroborates our proposed model of increased catalytic activity due to electronic structure changes of Pt NPs.

-d-band center calculations: Our p-DOS analysis revealed that the d-band center of small Pt nanoparticles (0.9 nm) differs markedly from those of larger nanoparticles (1.2

and 1.6 nm). This finding is consistent with both the Bader charge analysis and the results from STEM-EELS, further supporting the proposed electronic structure effect.

In summary, our findings unveil a striking transformation in the catalytic behavior of Pt nanoparticles as their size surpasses the critical 1–1.5 nm threshold. This transition is marked by a sharp shift in activation energy (DFT) and electronic structure (STEM-EELS and d-band center), highlighting a fundamental change in the nature of active sites. Our experimental catalytic data and spectral structure information quantitatively reflect these electronic and energetic transitions, exhibiting an excellent fit to the Arrhenius kinetic model ($R^2 = 0.95$). These results provide compelling evidence for the size-dependent electronic and kinetic properties of Pt catalysts.

Listed below are a few questions/comments for the authors to consider. There are many other technical questions. The reviewer, however, considers these as critical.

1) With particle sizes of ~ 1 nm, would the concept of "corner", "edge", or "terrace" atoms be still a valid description for the nature of the metal species? In this size range, their "shape" or configuration should strongly depend on the nature of the support surfaces. It is expected that the strong metal-support interactions dominate the metal configurations and modify their electronic structures as well. Both the heterogeneities on the support surfaces and the different sizes of the metal particles contribute to the "broadening" of the experimentally observed XPS spectra. From this perspective, one could not simply classify the active sites as "corner", "edge" and "terrace" sites on the metal particles. In fact, the so called "corner" sites should not be considered catalytically equivalent since the catalytic property of each individual "corner" site depends on the size of the metal particle and how that specific particle interacts with the support surface.

Thank you for raising these important concerns regarding our work. We have carefully addressed them as follows:

Morphology of Small-Sized NPs:

Firstly, it is important to emphasize that ~1 nm Pt NPs on CeO₂ exhibit well-defined structures. For instance, the aberration-corrected STEM images of Pt NPs on various CeO₂ surfaces, as shown in Figure R2-1, clearly depict NPs of approximately 1 nm in size, conforming to a truncated octahedral model (as detailed in Section S2.2) with distinct edge and corner atoms visible. Images of the Pt/CeO₂ catalyst under WGS

working conditions by in situ TEM also show well-defined Pt structure with the dynamics of corner perimeter atoms, as depicted in Figure R2-2.

Figure R2-1. STEM images of Pt/CeO₂ with approximately 1 nm Pt clusters.

[Figure Redacted]

Figure R2-2. Images of the Pt/CeO₂ catalyst and the schematics of the active structure of the Pt nanocluster/CeO₂ in WGS reaction conditions. This figure is reproduced from *Nat Commun* 12, 914 (2021).

Secondly, regarding Pt NPs smaller than 1 nm, we observed only a few of them in our STEM images, with the vast majority of NPs being larger than 1 nm. Therefore, our conclusions primarily pertain to NPs of approximately 1 nm or larger, and we have updated the corresponding TOC image accordingly. For NPs smaller than 1 nm, which are difficult to visualize in STEM images, the majority of atoms are indeed low-coordination corner or edge atoms, a point we have addressed in our peak fitting analysis in Section S2.1.

We have taken your suggestion into account to enhance the clarity of S2.1.

"In practical situations, for example, when AD begins to sinter into assemblies of a few atoms, it is challenging to identify and account for these smaller clusters using electron microscopy. Considering that these sub-nanometer clusters are primarily composed of edge and corner atoms, this leads to a higher proportion of corner atoms than the average visible in Pt NPs. Therefore, we added the component 'corner2' to fit the potential corners from smaller clusters. It has the same BE, FWHM, G/L ratio, and asymmetry as 'corner1', but the area is not constrained. 'Corner2' makes no contribution when there are large platinum particles, but it becomes prominent during structural evolutions, e.g., AD_1 at 250°C during ramp-up (Figure S12(B)). The corner sites will eventually be the sum of 'corner1' and 'corner2'.

Impact of Pt NPs Size on Pt 4f Spectra:

We acknowledge that the size of platinum NPs does indeed influence the Pt 4f photoemission spectra. Since this issue has been brought up again in the following comment (2), we will make sure to comprehensively discuss it in our forthcoming response.

Impact of Support Heterogeneity on Pt 4f Spectra:

As discussed in *Angew. Chem. Int. Ed.* 2022, 61, e202112640, the Pt 4f spectra for Pt deposited on CeO₂(111)-step surfaces are broader than those on well-ordered stoichiometric CeO₂(111) surfaces. This broadening occurs because the CeO₂(111) surface primarily features metallic Pt clusters, which contribute a peak at 71.8 eV. In contrast, the CeO₂(111)-step surface introduces monoatomic steps that allow for the binding of single Pt atoms. These single atoms create an additional peak at 72.8 eV,

thereby broadening the overall Pt 4f spectrum. The study also examined the effects of reducing CeO₂(111) to create oxygen vacancies, which influences Pt-CeO₂ interaction, but found that the Pt 4f spectra for Pt/CeO₂-red were similar to those of Pt/CeO₂(111).

Therefore, the impact of support heterogeneity on Pt structure is primarily reflected through changes in size, oxidation state, and morphology of Pt NPs, which in turn influence the spectral features. For example, in *Angewandte Chemie International Edition* (47: 2884-2887), the catalytic activity of Au/CeO₂ on different CeO₂ facets varies significantly, largely due to differences in nanoparticle size and oxidation state. Similarly, in *ACS Appl. Mater. Interfaces* 2024, 16, 33, 43556–43564, CeO₂ facets were found to determine the dispersion and size of Pt species under redox conditions.

On the other hand, Pt-CeO₂ interactions can exhibit variations across different CeO₂ facets, leading to distinctive shapes of Pt NPs. This phenomenon, as discussed in *Nanoscale* (2023, 15, 19091-19098), suggests that Pt may undergo flattening on the CeO₂(100) facet due to the stronger interaction between Pt-CeO₂(100) than Pt-CeO₂(111). We have discussed the flattened Pt model in Section S2.2 and Figure S15. However, due to the lower surface energy, CeO₂(111) facet has the highest stability, which is the most common crystal planes existing in nanoparticles (*Phys. Rev. B* **69**, 075401; *J. Am. Chem. Soc.* 2002, 124, 38, 11429–11439; *Angew. Chem. Int. Ed.*, 53: 12069-12072; *Applied Catalysis B: Environmental* 294 (2021): 120257.). Given the broad readership of Nature Communications, it is crucial to avoid overgeneralizing from specific cases. Our study intentionally avoided using special single crystals or preferred support facets (e.g., CeO₂(100)). Instead, we sought to represent the most statistically significant scenario. Based on the STEM images (Figure R2-1 and R2-2) and statistical analysis of small Pt NPs in Figure R2-3 (*Nanoscale Adv.*, 2022, 4, 3978-3986), the 3D Wulff model or truncated octahedral model provides the most appropriate representation of Pt NP distribution on CeO₂.

[Figure Redacted]

Figure R2-3. (A) The distribution of platinum particle shapes within each size range using the combination of four different synthesis methods. The colors correspond to

particle shapes using the following acronyms: truncated octahedron (TO), truncated cuboctahedron (TC), over-truncated cuboctahedron (OTC), and tetrahedron or truncated tetrahedron (T or TT). The statistics of the area fraction of each facet type (relative to the total surface area at a given size) from experiment **(B)**, and from Boltzmann statistics **(C)**. The figures were reproduced from *Nanoscale Adv.*, 2022, 4, 3978-3986.

Catalytically inequivalent corner sites due to electronic structure effects:

We fully agree with the point raised—this is actually the core message of our work. As we highlighted in the manuscript: "*However, Figure 2(H) shows that when the WGS activity is correlated with the fraction of corner sites, the relationship is not proportional. As an example, the AD_5 sample, which has a fraction of ~7% corner sites, displays a WGS rate of $0.00025 \text{ mol}_{\text{CO}} \cdot \text{mol}_{\text{Pt}}^{-1} \cdot \text{s}^{-1}$, whereas the rate rises to $0.017 \text{ mol}_{\text{CO}} \cdot \text{mol}_{\text{Pt}}^{-1} \cdot \text{s}^{-1}$ in the most active NP_0, which has 18% corner sites (The corner atoms increased by less than three times, yet the activity increased nearly a hundredfold). The potential explanation is that the corner sites are not all active to the same extent, suggesting that an additional electronic structure effect plays a role.*"

Building on this point, we further investigated how different Pt NP sizes influence catalytic activity. Our conclusion is that the electronic structure of Pt, which varies with NP size, determines the WGS reaction activity of different corner or perimeter sites.

2) The authors should provide details on the validity of using XPS spectra to determine the corner, edge, and terrace sites with respect to the particle sizes. Does this method work for any sizes of the Pt particles? Which Pt sizes are most appropriate for using such a deconvolution method?

Thank you for the extended attention to the XPS data processing. We will address the concerns thoroughly, including the issues raised in previous comment (1).

In general, as the size of Pt decreases, the centroid of Pt 4f tends to shift towards higher binding energies (*Nature Mater* 15, 284-288, 2016). Below we listed two examples where the sizes of Pt differ:

1. **Large Size:** According to the study in *Science* 327, 850-853, 2010, the binding energies for corner & edge, terrace, and bulk sites in Pt(557) are assigned at 72.2 eV, 71.6 eV, and 71.0 eV, respectively. In this case, Pt(557) represents a bulk sample, suggesting that the Pt size is relatively large. Even if we consider the

induced Pt cluster size, it would be around 2-3 nm, which is in alignment with the size of Pt NPs in our study.

2. **Small Size:** In contrast, in *Nature Mater* 15, 284-288, 2016, when Pt is gradually deposited on CeO₂, the smallest Pt clusters exhibit a Pt 4f_{7/2} binding energy of 72.0 eV. These clusters primarily consist of corner and edge atoms, a position close to the 72.2 eV discussed above in point 1. Given that these clusters are smaller than Pt(557), their binding energy should shift to an even higher energy. However, due to the energy resolution of XPS and the broadening effects caused by the presence of a few higher-coordination terrace/bulk atoms at lower binding energies, an average peak position of 72.0 eV is reasonable. This suggests that, for small Pt clusters, the binding energy for low-coordination corner and edge atoms should fall within the range of 72.0-72.2 eV.

Therefore, considering the above examples, the binding energy for corner atoms in both small clusters and larger NPs is approximately 72.2 eV. Given instrumental differences and work function variations, using 72.2 eV as the binding energy for low-coordination corner and edge atoms is reasonable and can be applied broadly across sub-nanometer to 3 nm Pt NPs.

Furthermore, as detailed in our main text and Supplementary Information, we have thoroughly considered the energy shift in Pt 4f spectra due to size effects. We compared the fitting results of the Pt 4f spectra and the kinetic model fitting before and after applying shift corrections. The results demonstrate satisfactory peak fitting and kinetic data fitting, for both shift corrected and uncorrected situations. For detailed results, please refer to Figures S21-S24 and Table S13. Additionally, we have revised the corresponding section in the main text page 13 to enhance clarity.

"Furthermore, it is necessary to consider how electronic structure effects influence Pt 4f spectra: a shift towards higher BE is counted on smaller NPs, particularly those smaller than 1.25 nm. For instance, the peak of terrace sites on small NPs (<1.25 nm) might shift to higher BE, i.e. from 71.6 to 72.0 eV (Lykhach, Kozlov et al. 2016). For edge/corner sites, the peak would also shift to higher energies (slightly higher than 72.2 eV but lower than 72.8 eV for AD species). Given the resolution of Pt 4f XPS measurements in this work (~0.3 eV) and the BE of small clusters on model Pt/CeO₂ (Lykhach, Kozlov et al. 2016), we assigned the peak at 72.2 eV to corner and edge sites, regardless of the NP size and of the potential BE shift of terrace atoms on small NPs. Table S7 lists the average diameter of small NPs (<1.25 nm) for all the samples and the corresponding fraction of corner, edge and terrace atoms. The fraction of corner sites

on small NPs is then corrected by removing the terrace*small contribution from the peak at 72.2 eV. The comparisons of fitting before and after terrace peak correction are shown in S2.6, suggesting that the correction has minimal influence on the data processing and the overall results."

3) At the bottom of page 6, the authors stated "The sintering of AD platinum into NPs activates the catalyst towards the low-temperature WGS reaction (Figure 1(F)). The majority of platinum remains dispersed (as AD species) and it probably contributes little to the catalytic activity." This is very confusing and misleading. The STEM image (Figure 1C) clearly shows that the AD species sintered to form Pt particles. There should not exist many AD species at all after the WGS. Even if the majority of the Pt species were AD species, then these species might have contributed significantly to the observed activity since these AD species were now reduced during the WGS process.

Thank you for your comments. It is important to acknowledge that the coexistence of AD and NPs on CeO₂ has been confirmed by XPS, XAS and AC-STEM results (*Angew. Chem. Int. Ed.* 2022, 61, e202112640; *J. Am. Chem. Soc.* 2016, 138, 48, 15743–15750; *J. Am. Chem. Soc.* 2020, 142, 1, 169–184), which is also one of the reasons for the ongoing debate regarding the active sites for WGS. As we have noted in the main text, despite using the most advanced electron microscopy, observing single Pt atoms on CeO₂ remains a significant challenge. Therefore, we attribute the absence of NPs to the solely presence of AD, but the presence of NPs does not negate the existence of AD. In this context, our primary method for detecting AD is XPS, a widely recognized and established technique for detecting and quantifying Pt AD species (*Science* 358, 1419–1423 (2017); *Nat Commun* 10, 1358 (2019); *Nat Commun* 7, 10801 (2016); *ACS Appl. Energy Mater.* 2018, 1, 12, 6781–6789; *ChemCatChem* 2024, e202301727; *Nat Commun* 14, 2664 (2023)). Regarding the reduced Pt single atoms, we have explicitly stated in our manuscript that under reduction conditions, the stability of single atoms is less favorable than the reduced nanoparticle clusters, leading to the aggregation of single atoms into nanoparticles. This observation is also supported by in situ XAS/ETEM (*Angew. Chem. Int. Ed.* 2017, 56, 13078.). Furthermore, reduced (metallic) platinum single atoms are thermodynamically unstable compared to the platinum nanoparticles, as also demonstrated by theoretical calculations (*Phys. Chem. Chem. Phys.*, 2020, 22, 28–38; *ACS Catal.* 2021, 11, 21, 13041–13049). While the literature has demonstrated the existence of reduced single atoms on specially prepared CeO₂ islands (Li, Pereira-Hernández et al. 2022), our powder samples do not employ such a synthesis strategy, making the occurrence of CeO₂ islands with isolated platinum atoms and ions highly

unlikely. Therefore, platinum in our samples is subject to the laws of thermodynamic stability, leading to the formation of nanoparticles as the predominant species.

On the other hand, the hypothesis of reduced single atom contributing to WGS is brought into question when considering that Pt 4f showed no energy shift of the AD species at 300°C (full reduction of single atom will cause the entire 72.8eV peak to move to lower binding energy). The emergence of minor spectral features at a lower binding energy is thus attributed to the NPs observed by STEM.

We have further stressed this point in the main text on Page 6.

The majority of platinum remains dispersed (as AD species) due to the major Pt 4f spectral features at a high binding energy and it probably contributes little to the catalytic activity.

4) Figure 1 caption: "(D) Experimental XPS spectra and peak fitting of the Pt 4f region obtained in 0.1 mbar CO + 0.3 mbar H₂O, during ramp-up at 100–300 °C and cool-down at 300–100 °C, using 620 eV of photon energy. Pt 4f region contains four components corresponding to bulk atoms (a), terrace atoms with bound CO (b), low-coordinated atoms with bound CO (c) and atomically dispersed (AD) Pt²⁺ ions (d)." If the authors' interpretation of the XPS spectra were correct, then most of the AD species did not sinter during the WGS process, contradicting the STEM imaging results and those in literature reports. If the most dominant peak in the XPS spectra (Figure 1d) does not originate from the AD species, then the rest of the analyses of the XPS spectra in this manuscript becomes irrelevant.

We noticed the continued engagement with AD identification, which (as mentioned in the previous comment) is extremely challenging also making use of the most advanced microscopy tools. We would like to clarify that our interpretation of the XPS data is consistent with both STEM observations and the existing literature. As we addressed in the previous response, the presence of NPs observed in STEM does not imply that all ADs have aggregated. Importantly, previous studies suggest that AD and NPs may co-exist during the WGS reaction. For example, in *Science* 350, 189–192, 2015, in-situ infrared spectroscopy clearly demonstrated that CO adsorbed on Pt single atoms does not participate in the WGS reaction, contrasting with the consumption of CO adsorbed on Pt NPs. This finding aligns with our view that AD does not serve as an active site. Consequently, even if AD is the dominant species on AD₀, its contribution to WGS activity is negligible.

5) Page 10, line 13. *"This is because XPS aids in accurately quantifying AD and NP species, thereby complementing the structural insights obtained from STEM."* This statement is not correct or at least misleading. There are no validated reports to show that XPS technique can be used to quantify the amount of AD species in a supported metal catalyst.

Thank you for raising this point, which like the previous pertains the assignment of XPS features. XPS is a widely recognized and established technique for detecting and quantifying Pt AD species (*Science* 358, 1419-1423, 2017; *Nat Commun* 10, 1358, 2019; *Nat Commun* 7, 10801, 2016; *ACS Appl. Energy Mater.* 2018, 1, 12, 6781–6789; *ChemCatChem* 2024, e202301727; *Nat Commun* 14, 2664, 2023). These studies often include complementary techniques such as HAADF-STEM and/or EXAFS for additional validation. In a previous work (*ACS Catal.* 2021, 11, 21, 13041–13049), we investigated the AD₀ catalyst by APXPS, XANES/EXAFS, HAADF-STEM and DFT calculations. AD platinum can be assigned to the feature at 72.8 eV in the Pt 4f_{7/2} peak, as also proven by X-ray absorption spectroscopy and electron microscopy.

We have modified the text on Page 10 to enhance the clarity:

The integration of XPS and STEM enables a comprehensive evaluation of the composition of sites present: XPS discriminates between AD and NP species while STEM provides the structure information of NPs to further deconvolute the number of particular Pt sites within them.

6) *In many places (main text and supporting information), it is not clear if the authors refer to general "corner" sites of a Pt nanoparticle or the specific "perimetric corner" sites of a supported Pt nanoparticle. These two different sites should be catalytically very different. If the authors refer to the "perimetric corner" sites, then some of the conclusions are not necessarily correct. For example, the number and nature of "perimetric corner" sites should be very different for "pancake-like" and "diamond-like" Pt nanoparticles. In fact, for smaller Pt particles, they usually do not show any clearly identifiable segments of perimeters. Therefore, the definition of "perimetric corner" sites becomes ambiguous or meaningless.*

Thank you for pointing out the potential ambiguity in our work. We would like to clarify the distinction here: perimeter corner atoms are a subset of corner atoms, specifically

those located at the Pt-CeO₂ interface, whereas part of corner atoms do not interact with CeO₂. We have applied these terms throughout the manuscript to enhance clarity.

Regarding the concern on the different perimeter corner fraction in the models of Pt NP, we have plotted the size dependence of the perimeter corner fraction for three shapes in Figure R2-4. As we discussed in the previous reply, the truncated octahedron is most statistically frequent shape and pancake model is the special case for stronger metal-support interaction on less common CeO₂(100) facet. The perimeter corner proportion between the truncated octahedron and pancake models does not differ significantly. Furthermore, as we have explained in our previous responses, for sub-nanometer clusters, the predominant species are low-coordination corner and edge sites. Therefore, we have made specific adjustments in our XPS peak fitting to account for these sites.

Figure R2-4. Size dependence of the perimeter corner fraction for three Pt NP models.

We have modified S2.1 to enhance the clarity:

In practical situations, for example, when AD begins to sinter into assemblies of a few atoms, it is challenging to identify and account for these smaller clusters using electron microscopy. Considering that these sub-nanometer clusters are primarily composed of edge and corner atoms, this leads to a higher proportion of corner atoms than the average visible in Pt NPs. Therefore, we added the component 'corner2' to fit the potential corners from smaller clusters. It has the same BE, FWHM, G/L ratio, and asymmetry as 'corner1', but the area is not constrained. 'Corner2' makes no contribution when there are large platinum particles, but it becomes prominent during

structural evolutions, e.g., AD_1 at 250°C during ramp-up (Figure S12(B)). The corner sites will eventually be the sum of 'corner1' and 'corner2'.

Comments in *blue* - Replies in black - Actions in *green*.

Reviewer #3

The authors have made a significant advancement in understanding water-gas shift (WGS) activity by integrating APXPS signal deconvolution results with STEM analyses to quantify the proportion of different Pt sites in the catalyst under varying conditions. This work establishes a quantitative relationship between WGS activity and specific Pt sites, advancing the widely accepted understanding of performance by pinpointing the critical role of corner atoms within the perimeter atoms, rather than just the general perimeter atoms. This progress, achieved through considerable time and effort, is highly commendable. However, rigorous language expression and analytical logic are essential in research to prevent catastrophic issues, and thus, the article requires further optimization in these points.

We sincerely appreciate your positive recognition of our work. Regarding the concerns raised about language expression and analytical logic in some parts of the manuscript, we partly agree with the Reviewer. Indeed, during multiple rounds of revisions, certain sentences inadvertently became ambiguous or misaligned in meaning. We have now carefully revised all sentences, pointed out and conducted an additional thorough review of the entire manuscript. This comprehensive editing ensures consistent and coherent language, logical clarity, and accurate scientific expression. We appreciate your insightful suggestions, which significantly enhance the rigor and readability of our manuscript.

1. On page 6, line 5, the article states, "Therefore, platinum in our samples is subject to the laws of thermodynamic stability, leading to the formation of NPs as the predominant species." This contradicts another statement: "The majority of platinum remains dispersed (as AD species)." Such contradictions, along with other instances of unclear expression, appear multiple times throughout the article, undermining its clarity and reliability.

We appreciate and agree with this comment. The manuscript has been modified accordingly as follows:

Page 6

Therefore, the reduction of Pt²⁺ is accompanied by the sintering of AD species into metallic NPs.

The interpretation of Pt being the major species after the WGS has been removed, as it would cause unnecessary confusion. Moreover, the reader can clearly recognize the information from Figure 1(D) and (E).

We further added more discussions about the Pt sintering and AD-NP coexistence on Pages 5-6.

At 300 °C, the originally symmetric Pt 4f spectrum becomes asymmetric, and can no longer be adequately fitted using a single-component model. This necessitates the introduction of an additional low-binding-energy component, which corresponds to metallic Pt⁰ nanoparticles formed upon partial reduction of Pt²⁺. This co-existence of AD and NP species is consistent with the post-reaction STEM observations shown in Figure S2.

2. In exploring the structure-performance relationship, the authors establish a linear correlation between small particle size and performance, focusing the study on this size range. Subsequently, a linear relationship between corner atoms on small particles and performance is identified. However, it is one-sided to assert, based on this analysis, that "This finding challenges the assumption that the corner sites

are equally active regardless of NP size, and that the mere geometric effect is responsible for the size effect of supported platinum for WGS".

Thank you for this valuable suggestion which further helps to improve the clarity of the discussion. We have revised the text accordingly to avoid overly assertive or potentially misleading statements.

Pages 13-14

"This finding reveals that the size-dependent effect of Pt/CeO₂ in the WGS reaction arises from two key factors: (1) an increased number of interfacial corner sites, and (2) an intrinsic activity enhancement of these sites when the particle size falls below a critical threshold."

"To quantify the intrinsic activity difference between CPS and other Pt sites, a non-linear Arrhenius kinetic model was developed and described in section S2.6, Figure S25 and Tables S10 to S12."

Reviewer #4

Comments to NCOMMS_504207: Quantifying Electronic Structure and Geometric Effects on the Activity of Pt/CeO₂ Catalysts for the Water-Gas Shift Reaction

This manuscript presents a comprehensive study of the structure–activity relationship in Pt/CeO₂ catalysts for the water-gas shift (WGS) reaction, combining operando X-ray photoelectron spectroscopy (XPS), scanning transmission electron microscopy (STEM), electron energy-loss spectroscopy (EELS), density functional theory (DFT) calculations, and kinetic modeling. The authors report that Pt atoms located at perimeter corner sites exhibit an intrinsic activity three orders of magnitude higher than other sites, primarily due to electronic structure effects, with a threshold particle size identified at approximately 1–1.5 nm. The manuscript is well-written, and the rebuttal letter thoroughly addresses the previous reviewers' comments.

However, I have several concerns regarding the robustness of the conclusions, particularly due to the heavy reliance on XPS fitting and the extrapolation of STEM data. These concerns warrant a major revision before the manuscript can be considered for publication.

We sincerely thank you for the thoughtful feedback and for recognizing the overall quality of our work. We greatly appreciate the constructive concerns raised regarding the robustness of our conclusions. In response, we have carefully addressed each of the comments in detail, provided new in situ/ex situ electron microscopy experimental data to strengthen the interpretation, and thoroughly revised the manuscript to improve clarity and analytical rigor. We hope that these efforts sufficiently address your concerns and bring the manuscript closer to meeting the publication standards of *Nature Communications*.

Major comments:

1. XPS Fitting and Site Deconvolution Accuracy:

The conclusion relies heavily on deconvoluting the Pt 4f spectra to assign site-specific signals (bulk, terrace, edge, corner, AD). However, the binding energy differences between these components—particularly edge, corner1, and corner2—are small (often <0.5 eV), and fall within the resolution limits of XPS. This overlap raises concerns about the reliability and reproducibility of site-specific quantification, especially when these subtle differences drive the key conclusions.

We appreciate your detailed evaluation and insightful comments regarding the reliability of the XPS peak deconvolution procedure. We fully understand the concerns about the small binding energy differences among edge, corner1, and corner2 components caused by their proximity to the resolution limit of the XPS technique.

To clarify, as shown in **Figure 1D**, the fresh AD₀ sample before activation exhibits a single asymmetric Pt 4f_{7/2} peak, which we attribute to atomically dispersed Pt species. Upon activation (e.g., AD₀ 300 °C), a clear low-binding-energy shoulder emerges that cannot be fitted by a single-component model. An additional component must be introduced to account for nanoparticle (NP) formation, a conclusion supported independently by STEM imaging of the post-WGS catalyst.

Thus, at the first level, the Pt 4f signal is deconvoluted into two main families: AD and NP. **Figure S15** (here presented as **Figure R2-1**) illustrates how these two components contribute to the overall envelope. The second step involves further resolving the NP component into site-specific

contributions—namely, bulk, terrace, edge, and corner sites. These site-specific components were introduced based on their physical origin and supported by previous literature (cited on Page 5 of the manuscript). Notably, the energy difference between bulk-terrace-edge & corner is **0.6 eV**, which can be well resolved by the overall resolution achieved making use of our synchrotron-based XPS setup (≤ 0.3 eV). We acknowledge that, due to the instrumental resolution, the binding energies of edge, corner1, and corner2 components may not be distinguishable as discrete peaks. As such, we have clarified in **Section S2.1** that these components were treated as a single peak with a shared binding energy, and their relative areas were determined by geometric modeling based on the particle size distribution extracted from STEM.

Importantly, all fitting parameters—including position, full width at half maximum (FWHM), line shape, and area ratios—were rigorously constrained across the entire set of collected spectra. This means that although multiple components were used in the deconvolution, they were not freely fitted; rather, any change in one component systematically influenced the others. In this way, the overall NP signal behaves as a tightly coupled multiplet, enhancing reproducibility and minimizing the risk of overfitting.

In summary, while the intrinsic resolution of XPS instrument imposes limitations on resolving edge and corner peaks, our analysis does not rely on resolving these components individually. Instead, it is based on a constrained fitting strategy grounded in physical structure models and validated by independent microscopy observations. We believe this approach provides a robust and reproducible framework for interpreting the site-specific evolution of Pt during WGS catalysis.

Figure R2-1. Peak fitting details of an example Pt 4f spectrum. (A) When fitting the raw data with AD and NP; (B) when continuing to divide NP into interrelated bulk, terrace, edge and corner; (C) when fitting the raw data with AD and interrelated bulk, terrace, edge and corner.

We have included the above discussion in the main text and SI.

Pages 5-6.

At 100 °C and 0% CO conversion, the Pt 4f spectrum can be fitted by a single symmetric component centered at 72.8 eV ($4f_{7/2}$), corresponding to AD species (i.e., Pt^{2+}). This agrees with the characteristic STEM image of AD species shown in **Figure 1(B)** and **S1**.

At 300 °C, the originally symmetric Pt 4f spectrum becomes asymmetric, and can no longer be adequately fitted using a single-component model. This necessitates the introduction of an additional low-binding-

energy component, which corresponds to metallic Pt⁰ nanoparticles formed upon partial reduction of Pt²⁺. This co-existence of AD and NP species is consistent with the post-reaction STEM observations in Figure S2.

S2.1

Although multiple components were used in the deconvolution, they were not freely fitted; rather, any change in one component systematically influenced the others. In this way, the overall NP signal behaves as a tightly coupled multiplet, enhancing reproducibility and minimizing the risk of overfitting. The main goal of using this framework, with its rigorous parameters and constraints, is to provide consistent and reliable interpretations of photoemission spectra.

While the intrinsic resolution of XPS instrument (~0.3eV) imposes limitations on resolving edge and corner peaks, our analysis does not rely on resolving these components individually. These components were treated as a single peak with a shared binding energy (72.2 eV for 4f_{7/2}), and their relative areas were determined by geometric modeling based on the particle size distribution extracted from STEM.

2. Use of STEM to Guide Global XPS Interpretation:

The authors use localized STEM images to inform the morphology and shape of Pt nanoparticles, which is then used to constrain XPS fitting across the entire catalyst. However, XPS is a global technique, averaging over a much larger sample area than STEM. The limited number of STEM images cannot reliably represent the full sample, raising questions about whether the assumed NP geometries and distributions are truly reflective of the bulk material.

Thank you for highlighting the spatial representativeness issue between localized STEM imaging and global XPS measurements. We fully acknowledge this intrinsic limitation: while STEM provides detailed structural information at the nanometer scale, it probes a far smaller area compared to the millimeter fraction-scale sampling area of XPS (the beam spot size is approximately 300 μm). Therefore, extrapolating particle morphology and distribution from a limited number of STEM images to guide global XPS analysis introduces an inherent uncertainty.

To address this concern, we have revised the manuscript to explicitly discuss this limitation and the precautions taken. Specifically, we included more images from multiple regions and low-magnification overviews that reveal broader particle distribution trends (see minor point 1 response). These efforts improve, though do not entirely resolve, the issue of spatial representativeness. Furthermore, our XPS fitting strategy was deliberately designed and parameter-constrained to ensure robustness even in the presence of local structural bias. For instance, in the AD_1 250°C spectrum, the fitted fraction of low-coordination Pt atoms from XPS is higher than the fraction estimated from post-reaction STEM guidance. This suggests the presence of ultrasmall particles not captured in the STEM survey, which are retrospectively inferred from the spectral data. Such reciprocal interpretation between spectroscopy and microscopy enables partial compensation for the limited spatial representativeness of STEM.

Taken together, the integrated analysis framework, combining complementary information from STEM and XPS with careful constraints management, provides a robust foundation for interpreting structural evolution within the catalyst. We also note, however, that improvements in spatial sampling and in situ size/shape estimate remain valuable directions for further refinement.

We have included the above discussion in SI as follows:

S2.1

We acknowledge the inherent spatial mismatch between STEM and XPS in terms of sampling volume. While STEM provides high-resolution structural information in the nm range, thus it is limited to local regions of the sample, XPS captures a global overview over a larger area (approximately 300 μm X-ray beam spot size). This difference introduces potential uncertainty when using localized STEM images to guide the fitting of photoemission peaks. To minimize this risk, the fitting strategy has been improved to ensure the robustness. In practical situations, when AD species start to sinter into assemblies of a few atoms, it is difficult to identify and account for these smaller clusters by electron microscopy. The proportion of small clusters may be higher, which leads to a higher proportion of corners. Therefore, we added the component "corner2" to fit the potential corners from smaller clusters. It has the same BE, FWHM, G/L ratio and asymmetry as "corner1", but the area is not constrained. "Corner2" has a negligible contribution upon AD species sintering into large platinum particles, but it becomes prominent during structural evolutions, e.g. AD_1 at 250°C during ramp-up (Figure S12(B)).

In this sense, global spectral fitting can also provide complementary information to microscopy, highlighting the synergistic potential of combining both techniques.

S2.2

As discussed in Section S2.1, nanoscale STEM images inherently carry localized information, whereas photoemission spectra provide area-averaged data in the hundreds of micrometer scale. To minimize potential spatial resolution bias, we acquired STEM images from multiple regions, including low magnification overviews in Figures S1-2, to better reflect the overall distribution and morphology of Pt NPs across the sample.

3. STEM Conditions vs. Operando Environment:

Moreover, the STEM data were acquired under high vacuum, whereas the XPS data were collected under WGS reaction conditions (gas flow and elevated temperature). Pt NP morphology and surface structure can change significantly under gas environments. While the authors cite prior in situ TEM studies and present ex situ STEM images, the referenced in situ work used different gas compositions and temperatures. These conditions may not accurately reflect the current system. The structure of Pt NPs was reported to go through very dynamic changes under reaction conditions, such as in the following recent papers: <https://www.nature.com/articles/s41467-021-26047-8>, <https://www.science.org/doi/10.1126/science.ads2688>, <https://www.nature.com/articles/s43246-024-00575-4#Sec13>. For greater confidence in the particle shape assignment under relevant conditions, in situ TEM data under the same WGS conditions used in this study would be highly valuable.

We appreciate this important comment and fully agree that observing nanoparticle morphology under reaction-relevant conditions (in situ) is essential for interpreting operando XPS results, especially given the potential fluxional nature of Pt nanoparticles and their susceptibility to destabilization. In response, we have made our best effort to conduct and provide two sets of additional in situ gas-cell STEM experiments, which demonstrate both the stability and the retention of the Winterbottom-like shape of the Pt nanoparticles under reaction conditions. Due to instrumental limitations, direct introduction of steam (H_2O) into the STEM gas-cell holder was not feasible. Therefore, we employed the reverse water-gas shift (RWGS) reaction ($\text{CO}_2 + \text{H}_2$), which produces CO and H_2O (*Appl. Catal. B: Environ.* 2021 291, 120107), as the closest experimentally accessible proxy to the actual WGS environment. Under this

condition, the catalyst is exposed to a mixture of $\text{CO} + \text{H}_2\text{O} + \text{CO}_2 + \text{H}_2$, which is chemically representative of WGS-relevant conditions. As further corroborated in Figure S43-S44, the Pt structural response, determined by XPS under $\text{CO} + \text{H}_2\text{O}$ exposure is equivalent to that observed under the $\text{CO} + \text{H}_2\text{O} + \text{CO}_2 + \text{H}_2$ mixture. These findings validate the use of the RWGS environment as a reasonable and physically relevant approximation of the WGS reaction conditions for in situ STEM investigation.

(1) In situ STEM on our real-world catalyst (NP_0):

We performed in situ STEM on the NP_0 sample using a gas-cell holder (DENSSolutions, Climate GVB) under controlled reaction environments (0.2 mbar $\text{CO}_2 + 0.2$ mbar H_2 at 250°C and 0.25 bar $\text{CO}_2 + 0.75$ bar H_2 at 300°C). The behavior of Pt NPs is presented in Movies S1-S2 and Figure R2-2. Due to the porous, high-surface area nature of the real-world catalyst, the projection images are complex—electron beams pass through multiple voids and particle layers, leading to overlapping contrast and reduced spatial resolution, especially under gas-cell conditions. Atomic-resolution imaging of the Pt NPs structure was challenging. Nevertheless, we were able to capture real-time movies of Pt NPs at 0.8s per frame, revealing that they maintain their morphology and show no apparent structural collapse or redispersion over several minutes. We acknowledge that higher temporal resolution may capture finer dynamics, but our acquisition rate is directly relevant to the XPS timescale: each Pt 4f spectrum was integrated over a ~ 10 minutes timespan, effectively averaging out transient dynamics. Thus, our STEM data demonstrate that, on this comparable timescale, the NP structures remain **stable** under mbar- and bar-level atmospheres.

Figure R2-2. Images of the NP_0 catalyst at 250°C in 0.2 mbar $\text{CO}_2 + 0.2$ mbar H_2 (A-C) and at 300°C in 0.25 bar $\text{CO}_2 + 0.75$ bar H_2 (D-F).

(2) Complementary in situ STEM on a model Pt/CeO₂ catalyst:

To focus on the shape of Pt nanoparticles at higher resolution, we provide in situ STEM results on a model 1% Pt/CeO₂ catalyst (*Small Methods* 2025, 9, 2401108) under 0.25 bar CO₂ + 0.75 bar H₂ at 300°C in Movie S3 and Figure R2-3. The high-resolution STEM images demonstrate that Pt nanoparticles retain a Winterbottom-like shape under the reaction conditions, undergoing only minor rotational motions without redispersion, collapse, or other structural changes that would strongly influence Pt 4f peaks deconvolution. These observations are consistent with previous reports showing that strong CO adsorption can transiently disturb surface Pt bonding, but once the WGS reaction proceeds (co-feeding H₂O), the particles stabilize into well-defined morphologies (*Nat Commun* **12**, 914 (2021)).

We acknowledge that the time resolution (~0.8 s/frame) of our experimental setup is lower than that of the sub-second studies cited (e.g. *Science* **387**, 949-954(2025); *Nat Commun* **12**, 5789 (2021); *Nat Commun* **12**, 914 (2021)), but our imaging timescale is nonetheless appropriate for understanding structural dynamics averaged over XPS acquisition times. In essence, the XPS data reflect a time-averaged signal over many transient states, and our in situ STEM results confirms that Pt NPs exhibit only limited configurational fluctuation within this timeframe. Therefore, although Pt NPs are inherently dynamic, the average structure over the course of minutes—as observed in both in situ STEM and XPS—is reliably described by the Winterbottom-like morphology. These findings support the validity of our structural assignment and modeling approach under WGS conditions.

Figure R2-3. Images of the model Pt/CeO₂ catalyst at 300 °C in 0.25 bar CO₂ + 0.75 bar H₂ (A-D) and the corresponding Fast Fourier Transform (FFT) of the Pt NP region. Panel E and G also show the FFT of ideal corresponding atomic models.

In short, although our in situ experiments were not conducted under precisely the same gas mixture as that used for XPS, we emphasize that we have implemented the closest achievable and chemically relevant conditions, and that both experiments provide strong evidence of a stable Winterbottom-like shape of the Pt NPs on the same timescale as that used for XPS scans.

We have added the above results and discussions on Pages 9-10 and S1.6 and S2.2. The recommended references have been cited and discussed.

Pages 9-10.

However, obtaining statistically significant shape distributions proves challenging due to various factors, including the reaction atmosphere, beam effect, image resolution, fluxional nanoparticle structure etc. Taking all these factors into account, we have selected the truncated octahedral shape as a meaningful and reliable representation based on theoretical calculations and in situ STEM evidence (a more comprehensive discussion refers to section S2.2).

S1.6 In situ gas-cell STEM and STEM-EELS Experiments

In situ gas-cell STEM experiments were performed on the DENSolutions Climate GVB system. Powder catalyst (NP_0 fresh) was loaded into a micro-electro-mechanical system (MEMS)-based nanoreactor. The catalysts were initially recalcined in the air (500°C, 1 bar) in the microscope to avoid carbon contamination, then cooled to RT. Subsequently, reaction mixture (0.2 mbar CO₂ + 0.2 mbar H₂ + 200 mbar N₂, 0.05 mL min⁻¹) was introduced and the nanoreactor was ramped up to 500°C then cooled down to 250°C for data acquisition. The NP_0 fresh sample and a model Pt/CeO₂(Eliasson, Niu et al. 2024) was also measured in higher pressure reaction mixture (CO₂: H₂=1:3, 1 bar, 0.3 mL min⁻¹) at 300°C. Images were acquired at a frame time ~0.8 s with the approximately dose rate 500 e⁻/Å²·s (spot size = 8). The electron beam was blanked when images were not being acquired. The time-resolved images were temporally binned by summing 5 frames, producing time-averaged images with an increased signal-to-noise ratio.

S2.2

As discussed in Section S2.1, nanoscale STEM images inherently carry localized information, whereas photoemission spectra provide area-averaged data in the hundreds of micrometer scale. To minimize potential spatial resolution bias, we acquired STEM images from multiple regions, including low magnification overviews in Figures S1-2, to better reflect the overall distribution and morphology of Pt NPs across the sample. On the other hand, platinum NPs can dynamically change their configurations under the reaction conditions. Recent advances in in situ electron microscopy with deep denoising reveal that upon CO exposure, Pt particles gradually deviate from their thermodynamically favored Winterbottom-like shape(Crozier, Leibovich et al. 2025). This morphological evolution is likely driven by local variations in Pt surface energy induced by Pt-CO strong interaction and changes in CO coverage, resulting in highly mobile atoms/clusters(Li, Kottwitz et al. 2021, Vincent and Crozier 2021). During O₂-H₂ pretreatment, surface atom migration on the catalyst has been observed(Li, Zakharov et al. 2024). Under CO oxidation reaction conditions, in situ TEM images display pronounced motion artifacts and features attributable to particle mobility(Vincent and Crozier 2021). However, under saturated CO coverage, the Pt NPs undergo reconstruction and still exhibits a truncated octahedron shape(Avanesian, Dai et al. 2017). In inert N₂ atmosphere, the dynamic behavior of Pt is significantly suppressed compared to that in CO(Vincent and Crozier 2021). Notably, under typical water-gas shift conditions (i.e., co-feeding H₂O), Pt nanoparticles become considerably more stable, with only peripheral atoms near the cluster edge exhibiting mobility(Li, Kottwitz et al. 2021).

To demonstrate the Pt NPs shape and stability under relevant conditions, we performed in situ STEM imaging of the powder catalyst sample (NP_0) using a gas-cell holder (DENSsolutions Climate GVB) under two different sets of CO₂ + H₂ environments: 0.2 mbar CO₂ + 0.2 mbar H₂ at 250 °C and 0.25 bar CO₂ + 0.75 bar H₂ at 300 °C, as shown in Figure S13 and Movies S1–S2. These conditions were chosen to bracket the pressure range from milli- to sub-bar levels, thus allowing the assessment of Pt NP behavior across relevant catalytic regimes. Due to the highly porous and high-surface area nature of the catalyst, projection images acquired under gas-cell conditions exhibit complex contrast: multiple voids, pores, and particle layers contribute to signal overlap and Z-contrast variation. As a result, direct atomic-resolution imaging of Pt was not achievable in these experiments. Nevertheless, dynamic sequences were successfully recorded at a frame rate of ~0.8 seconds, allowing us to track the temporal evolution of Pt nanoparticles with sufficient temporal resolution for reaction-relevant interpretations. Across both pressure regimes, the Pt nanoparticles maintained their size and morphology over several minutes of continuous imaging, showing no evidence of redispersion, sintering, or collapse. Although it is possible that finer structural dynamics (e.g., surface atom migration) may occur above our temporal and spatial resolution, the observed stability strongly supports the notion that on the 1–10 minutes timescale relevant for operando XPS (e.g., Pt 4f acquisition typically spans ~10 min), the NPs exist in a structurally stable state. This set of experiments also confirms that Pt NP behavior is qualitatively consistent across mbar- to sub-bar- pressure conditions. Despite limitations in spatial resolution, these real-time observations provide crucial experimental evidence that the structural assignment used in XPS interpretation—based on stable NP configurations—is physically meaningful under realistic gas environments.

Figure S14 and Movie S3 present a time-resolved series of atomic-resolution HAADF-STEM images (A–D) of a Pt nanoparticle supported on CeO₂, along with the corresponding Fast Fourier Transforms (E–H). These snapshots, acquired at 00:00, 00:35, 01:42, and 02:50, respectively, allow direct visualization of the NP's orientation dynamics during observation. At t=00:00 (A), the particle displays a projected morphology with nearly hexagonal symmetry and well-defined lattice fringes, as indicated by the overlaid structural model. The corresponding FFT (E) reveals features that approximate a threefold symmetry, which resembles that of a [111] zone axis in face-centered cubic (fcc) Pt. While neither the real-space image nor the FFT exhibits a perfect [111] projection, this orientation serves as a reasonable approximation. At t=00:35 (B) and more clearly at t=01:42 (C), the particle undergoes a significant rotational change, and the originally visible atomic columns become indistinguishable. The corresponding FFT pattern (G) also evolves from a nearly threefold symmetric pattern into a rectangular distribution, indicating a deviation from the [111] zone axis. The FFT shows features that are more characteristic of a projection close to the [101] zone axis. The overlaid structural model and its FFT further support this interpretation. Although the nanoparticle is not perfectly aligned with the [101] zone axis, the observed image and FFT features are consistent with a reorientation from [111] to [101] (tilting angle between [111] and [101] = 35.26°). At t = 02:50 (D), the diffraction pattern (H) stabilizes, and the real-space fringes no longer evolve, indicating that the particle has reached a new steady-state configuration. Throughout the sequence, the particle retains a truncated-octahedral-like shape without signs of sintering, facet disorder, or significant atomic rearrangement. This sequence demonstrates a structurally coherent and directional rotation of the Pt NP from [111] to [101], without morphological collapse. The transformation is likely driven by interfacial energy minimization or interaction with the CeO₂ surface. Importantly, this reorientation appears to occur via a rigid-body rotation rather than surface diffusion or reconstruction.

Figure S13. Images of the NP_0 catalyst at 250 °C in 0.2 mbar CO₂ + 0.2 mbar H₂ (A-C) and at 300 °C in 0.25 bar CO₂ + 0.75 bar H₂ (D-F).

Figure 14. Images of the model Pt/CeO₂ catalyst at 300 °C in 0.25 bar CO₂ + 0.75 bar H₂ (A-D) and the corresponding Fast Fourier Transform (FFT) of the Pt NP region. Panel E and G also show the FFT of ideal atomic models.

Minor Points:

1. Figure 1 – AD vs NP Claim:

The claim that only AD Pt is present before WGS, and only aggregated NPs are observed afterward, is not fully supported by the current STEM images in Fig. 1 and Fig. S5. The authors should provide images at same magnification at both atomic scale and lower magnification to show representative regions of the CeO₂ support before and after reaction.

We respectfully disagree with this interpretation. Our claim is not that all atomically dispersed Pt species aggregate into nanoparticles after WGS, but rather that a fraction of the AD species aggregate into NPs while a significant portion remains atomically dispersed. This coexistence of AD and NP species is supported by both STEM and XPS data.

To clarify this point, we have now provided additional STEM images acquired at both atomic resolution and low magnification to illustrate representative regions before and after reaction. As shown in Figure S1, no NPs are observed in the fresh sample, even at low magnification. However, after the WGS reaction in Figure S2, low-magnification overview images clearly reveal the formation of Pt NPs, while high-resolution STEM still detects atomically dispersed Pt species. This directly supports our claim that AD and NP species coexist after reaction, rather than suggesting a complete conversion of AD into nanoparticles.

Furthermore, the XPS results provide additional, independent evidence of this AD/NP coexistence. In the fresh sample, the Pt 4f spectrum shows a single symmetric peak centered at 72.8 eV, characteristic of atomically dispersed Pt. After the WGS reaction at 300 °C, this peak becomes asymmetric, with an emerging shoulder at lower binding energy. This new component corresponds to metallic Pt in NP form, while the main component remains centered at 72.8 eV, indicating that a large fraction (>70%) of AD species still exists after the reaction.

We have revised the manuscript to clarify this interpretation and included the new STEM images as suggested. These additions clearly illustrate the structural evolution of Pt species and confirm the coexistence of AD and NP species after WGS, consistent with the spectroscopic and microscopic evidence.

Pages 5-6

At 100 °C and 0% CO conversion, the Pt 4f spectrum can be fitted by a single symmetric component centered at 72.8 eV (4f_{7/2}), corresponding to AD species (i.e., Pt²⁺). This agrees with the characteristic STEM image of AD species shown in **Figure 1(B) and S1**. No change in the Pt 4f spectrum occurs upon a temperature increase to 250 °C. At 300 °C, the originally symmetric Pt 4f spectrum becomes asymmetric, and can no longer be adequately fitted using a single-component model. This necessitates the introduction of an additional low-binding-energy component, which corresponds to metallic Pt⁰ nanoparticles formed upon partial reduction of Pt²⁺. This co-existence of AD and NP species is confirmed by the post-reaction STEM observations in Figure S2.

AD_0-fresh

AD_1-fresh

AD_5-fresh

Figure S1. Representative STEM images of the fresh AD_x catalysts.

Figure S2. High resolution HAADF-STEM images of used AD₁ at various magnification levels, presenting the co-existence of AD and NP species.

2. EELS – Beam-Induced Reduction of CeO₂:

In Fig. S31, EELS spectra are only shown for regions near the Pt/CeO₂ interface. Since electron beams can reduce Ce⁴⁺ to Ce³⁺, a reference EELS spectrum from bulk CeO₂ under identical acquisition conditions is needed to confirm that the observed changes are due to Pt–support interaction rather than beam effects.

Thank you for this valuable suggestion. To address this concern, we have now included additional EELS spectra acquired under identical beam conditions from a bulk region of CeO₂ without nearby Pt particles. As shown in the updated Figure S35 (the previous Fig. S31), the spectral features of bulk CeO₂ remain clearly distinct from those near the Pt/CeO₂ interface. Specifically, the Ce M_{5,4} edge in the bulk region maintains a profile characteristic of Ce(IV), with no noticeable shift nor intensity change indicative of reduction to Ce(III). In contrast, the regions adjacent to Pt NPs show evident spectral changes consistent with partial reduction.

These observations confirm that the changes in Ce oxidation state near Pt are not caused by beam-induced effects, but rather reflect genuine Pt–support interactions. We have updated the figure and corresponding text in the Supporting Information to highlight this distinction.

Figure S35. HAADF-STEM image of model Pt/CeO₂ catalyst with different size of Pt NPs (A) 1.4 nm and (B) 1.7 nm. The labels indicate the EELS acquisition position. (C) An example of background subtraction for Ce M_{5,4} edge core loss spectrum. (D) Ce M_{5,4} edge EEL spectra of the labelled position on CeO₂ in the vicinity of small and large Pt NPs and bulk CeO₂ after background correction.

3. EDX Data Quality – Fig. S36:

The EDX spectra shown have low counts, making it unclear whether the absence of Na peaks reflects true absence or insufficient acquisition time. The authors should improve the EDX signal quality or clarify the capture parameters used.

We appreciate this feedback on the EDX data quality. We have repeated the EDX experiment on the same sample under optimized conditions. The new results confirm our original conclusion: sodium is homogeneously distributed throughout the catalyst, without any significant enrichment near Pt nanoparticles or near atomically dispersed Pt species.

The EDX mapping was performed on a FEI Titan Themis operated at 300 kV in STEM mode using a SuperEDX system. The experiment was conducted with a beam convergence of 18.0 mrad, a screen current of approximately 80 pA. Spectrum images were acquired with a pixel resolution of 0.051 nm/pixel and a dwell time of 40.0 μ s/pixel. A total of 30 frames were recorded and aligned using frame-by-frame drift correction to enhance signal-to-noise ratio.

Figure S40. STEM and EDXS images of used AD_1. Homogeneous dispersion of sodium is observed in two representative regions containing Pt NPs and Pt ADs.

4. Gas Pressure Clarification – Page 4, Lines 93–94 and Fig. 1A:

The feeding composition is given as 100 ppm CO and 300 ppm H₂O, but the total pressure is not specified. Please clarify the actual pressure used in these flow reactor experiments.

The actual total pressure is 1 bar and the corresponding partial pressures of 100 ppm CO + 300 ppm H₂O are 0.1 mbar CO + 0.3 mbar H₂O. This is further specified in S1.2.

S1.2

The flow mixture was 100 ppm CO, 300 ppm H₂O and argon to balance, at a total flow rate of 100 ml/min in ambient pressure.

5. Microscope Specification – Methods Section:

Instead of broadly stating “aberration-corrected TEM,” please specify whether the instrument used was probe-corrected or image-corrected to clarify imaging capabilities. In S1.6, Line 1129, “the spot size was adjusted to keep the beam current low in order to minimize beam-effects”, please specify the spot size and dose rate.

Thank you for the suggestion. To clarify, all atomic-resolution STEM imaging in this study were performed using probe-corrected microscopes, i.e. Titan Themis and Hitachi HD-2700.

Regarding the beam condition, we used a spot size of 7 and the dose rate per frame was approximately $2.4 \times 10^4 \text{ e}^-/\text{\AA}^2\text{-s}$, as calculated from the measured screen current. These specifications have been added to the revised S1.3 and S1.6 for clarity.

S1.3

Scanning TEM (STEM) at 200 kV, with high-angle annular dark field (HAADF) imaging, was performed with a Hitachi HD 2700 electron microscope, equipped with a probe-corrector, at the Scientific Center for Optical and Electron Microscopy (ScopeM) of the ETH Zurich.

S1.6

For in situ STEM: Images were acquired at a frame time $\sim 0.8 \text{ s}$ with the approximately dose rate $500 \text{ e}^-/\text{\AA}^2\text{-s}$. The electron beam was blanked when images were not being acquired.

For ex situ STEM: The analysis of the sample was conducted at room temperature under vacuum using a probe-corrected Titan Themis microscope operating at 300 kV. The beam convergence angle was set to approximately 26 mrad, with the approximately dose rate $2.4 \times 10^4 \text{ e}^-/\text{\AA}^2\text{-s}$.

Recommendation:

I recommend major revision to address the concerns above. While the manuscript presents a novel and thorough investigation into the electronic structure effects of Pt/CeO₂ catalysts, the key conclusions rest on assumptions that require more robust experimental validation. Addressing the issues related to XPS–STEM correlation and in situ structural fidelity will significantly enhance the strength and credibility of the work.

We sincerely thank you for the thorough evaluation and positive comments on our work. We fully understand the concerns regarding the correlation between XPS and STEM analyses, as well as the structural fidelity under in situ conditions. We agree that these issues are critical for enhancing the reliability of our conclusions. As detailed in our point-by-point responses above, we have made substantial revisions to the manuscript and incorporated additional experimental data to address the concerns raised. We believe that these additions and clarifications have significantly strengthened the

experimental foundation and further improved the robustness of our interpretations. Once again, we greatly appreciate your insightful comments, which have helped us improve the rigor and clarity of the manuscript.

Comments in *blue* - Replies in black - Actions in *green*.

Reviewer #4

In the revised manuscript, the authors have adequately addressed the concerns I raised in the initial review. The updates enhance the presentation and improve the manuscript's clarity and coherence. I commend the authors for their responsiveness. I recommend to accept this paper.

Thank you for your positive feedback and for recognizing the quality of our work. We worked hard to address all comments and we agree that the manuscript quality has improved.

One minor point is about the in-situ STEM data, in the labeled time stamp '02:50', the time format should be clarified — is this in mm:ss or hh:mm? Specifying the format would enhance clarity for the reader.

Thank you for noticing this. The time format (mm:ss) is now specified in the captions of in situ STEM data.

The captions of Supplementary Movies have been updated as follows:

- Supplementary Movie 1. Dynamics of NP₀ catalyst in 0.2 mbar CO₂ + 0.2 mbar H₂ at 250 °C. The labeled time stamp is in mm:ss.
- Supplementary Movie 2. Dynamics of NP₀ catalyst in 0.25 bar CO₂ + 0.75 bar H₂ at 300 °C. The labeled time stamp is in mm:ss.
- Supplementary Movie 3. Dynamics of the model Pt/CeO₂ catalyst at 300 °C in 0.25 bar CO₂ + 0.75 bar H₂. The labeled time stamp is in mm:ss.